# Diverging co-translational protein complex assembly pathways are governed by interface energy distribution

Johannes Venezian [1], Hagit Bar-Yosef[1], Hila Ben-Arie Zilberman[1], Noam Cohen[1], Oded Kleifeld [1], Juan Fernandez-Recio [2], Fabian Glaser[3] & Ayala Shiber [1] ✉

Protein-protein interactions are at the heart of all cellular processes, with the ribosome emerging as a platform, orchestrating the nascent-chain interplay dynamics. Here, to study the characteristics governing co-translational protein folding and complex assembly, we combine selective ribosome profiling, imaging, and N-terminomics with all-atoms molecular dynamics. Focusing on conserved N-terminal acetyltransferases (NATs), we uncover diverging co-translational assembly pathways, where highly homologous subunits serve opposite functions. We find that only a few residues serve as "hotspots," initiating co-translational assembly interactions upon exposure at the ribosome exit tunnel. These hotspots are characterized by high binding energy, anchoring the entire interface assembly. Alpha-helices harboring hotspots are highly thermolabile, folding and unfolding during simulations, depending on their partner subunit to avoid misfolding. In vivo hotspot mutations disrupted co-translational complexation, leading to aggregation. Accordingly, conservation analysis reveals that missense NATs variants, causing neurodevelopmental and neurodegenerative diseases, disrupt putative hotspot clusters. Expanding our study to include phosphofructokinase, anthranilate synthase, and nucleoporin subcomplex, we employ AlphaFold-Multimer to model the complexes' complete structures. Computing MD-derived interface energy profiles, we find similar trends. Here, we propose a model based on the distribution of interface energy as a strong predictor of co-translational assembly.

The majority of proteins do not function alone; rather, they form intricate cellular networks. Current research estimates that ~60% of the proteome forms stable, hetero-oligomeric protein complexes in the model eukaryote *Saccharomyces cerevisiae*[1,2]. Despite how ubiquitous these complexes are, the mechanisms underlying complex assembly remain largely obscure; Surprisingly little is known about the mechanisms ensuring the efficiency and specificity of protein-complex assembly. Many in vitro and in silico studies suggest assembly is driven by diffusion and random collisions of partner subunits[3,4]. This mechanism is problematic, as in the crowded environment of the cell,

non-productive interactions can out-compete the productive ones, particularly among subunits with either a low diffusion rate, low abundance, and high aggregation or degradation propensities[5]. Supporting this claim, a recent study demonstrates that co-translational assembly is prevalent in *S. cerevisiae*[6], with the importance of co-translational assembly further underscored by the high aggregation propensity of orphan subunits, which cannot fold independently (e.g., chromosome abnormalities such as Copy Number Variations, a hallmark of cancer genomes, often lead to the aggregation of orphaned subunits[7]). Out of the twelve complexes analyzed, nine were identified

[1]Faculty of Biology, Technion Israel institute of Technology, Haifa, Israel. [2]Instituto de Ciencias de la Vid y del Vino (ICVV), CSIC-Universidad de La Rioja-Gobierno de La Rioja, Logroño, Spain. [3]Lorry I. Lokey Interdisciplinary Center for Life Sciences & Engineering, Haifa, Israel. ✉e-mail: ayalashiber@technion.ac.il

to have dedicated subunits binding their partner subunit in the complex as they emerge from the ribosome exit tunnel. These dedicated subunits harbor a function beyond their enzymatic activity, directing the folding and assembly of the entire complex in a chaperone-like manner[6]. Furthermore, co-translational assembly was shown to orchestrate competing biogenesis pathways and prevent promiscuous interactions[8–13]. However, mechanistic characterization of assembly pathways remains a major challenge, as diffusion, recognition, and binding are all coupled to the dynamic processes of synthesis and folding.

To unravel the underlying mechanism by which subunits associate and fold co-translationally, we studied two highly conserved complexes belonging to the N-terminal acetyltransferases (NAT) family: NatA and NatB. These NATs form stable heterodimers, composed of a catalytic and an auxiliary subunit, that catalyze the addition of an acetyl group to various proteins as they emerge from the ribosome[14]. Each complex has distinct substrate specificities, determined by the N-terminal amino-acid sequence[15–17]. N-terminal (Nt)-acetylation is one of the most prevalent protein modifications in eukaryotes and is involved in vascular, hematopoietic, and neuronal growth and development[18–21]. The NATs show a very high structural and sequential homology[22] and offer a compelling native model system to study folding and assembly pathways. Several studies suggest the NATs' folding pathways are highly susceptible, even to single mutations with multiple disease-related single nucleotide polymorphisms (SNPs) identified as leading to the synthesis of unstable subunits, which fail to assemble and show high aggregation propensity[23–25]. These SNPs are associated with various cancers and developmental syndromes[26–29]. Even though the subunits exhibit a very high structural and sequential homology, the two complexes harbor opposite co-translational assembly pathways discovered by selective ribosome profiling (SeRP)[6,30].

To study the differences in the mode of co-translational assembly and its directionality, we perform SeRP complimented with multiple molecular dynamics (MD) simulations at different temperatures. We identify critical amino acids as well as structural and energetic features crucial for complex formation and stabilization. The MD predictions are then tested and validated in vivo in *S. cerevisiae* by RNA Immunoprecipitation (RIP)-qPCR, growth assays, imaging, and N-terminomics. Our investigation expands to include additional complexes, such as phosphofructokinase, anthranilate synthase, and nucleoporin subcomplex. We employed AlphaFold-Multimer[31], then MD-derived interface energy profiles, showing similar trends. Our findings reveal the predictive power of energy profiling in identifying anchor residues critical for co-translational interactions. Finally, a model for co-translational complex assembly prediction and its direction is proposed.

## Results

### NatA and NatB diverging co-translational assembly pathways correlate with opposite stability propensities

To elucidate the mechanistic aspects governing co-translational assembly pathways, we focused on two highly homologous complexes: NatA and NatB (Fig. 1a). These heterodimeric complexes offer a compelling native model system to study folding and assembly pathways, as disruption of complexation leads to neurodevelopmental disorders[18,19,24–26,32,33]. All the NAT complexes from yeast to humans share high sequence homology and an analogous domain organization[14,22,34,35]. Our previous study indicated that many complexes, including the NATs, assemble co-translationally[6]. These results prompted us to further investigate the NATs assembly pathways.

To capture co-translational assembly interactions of the NATs in vivo, we generated four *S. cerevisiae* strains. Each strain chromosomally encodes one of the complexes' subunits C-terminally fused to EGFP for immunoprecipitation (IP). SeRP compares the distribution of ribosome-protected mRNA footprints of two distinct samples generated from a single culture. One comprises the ribosome-protected footprints of all translated open reading frames (ORFs) (total translatome). The other contains footprints of a selected set of ribosomes, co-purified with a tagged interaction partner (interactome). Enrichment of footprints in the interactome, as compared to the total translatome, directly reports on co-translational interactions[6]. We validated that tagging does not affect any of the subunits' function by growth assay analysis under optimal conditions and mild heat stress, where both NatA and NatB null mutants show growth inhibition (Supplementary Fig. 1a). Using SeRP, we analyzed the co-translational interactions of each of the NATs' subunits. Remarkably, while both complexes exhibit co-translational assembly (Fig. 1b), these homologous complexes follow opposite assembly pathways[6]. NatA assembly initiates with the engagement of the nascent auxiliary subunit (Naa15) by the fully folded catalytic subunit (Naa10), while the auxiliary subunit does not engage ribosomes translating its partner. In contrast, the assembly of NatB initiates by the fully translated auxiliary subunit (Naa25) engaging the nascent catalytic subunit (Naa20) while its catalytic partner does not engage its partner co-translationally (Fig. 1b). Both co-translational assembly interactions are substantial, leading to at least 200-fold enrichment of interactome-to-translatome ribosome-protected footprints. For the NatA complex, the onset of co-translational interaction was observed after 417 residues of the auxiliary subunit were exposed from the ribosome exit tunnel. Thus, amino acids (aa) 1-417 represent the minimal region required to form a stable complex association. In contrast, for NatB, the minimal region of the catalytic subunit required to form a stable complex association on the ribosome is comprised of the first 104 aa (Fig. 1b).

To find the structural determinants of these opposite assembly pathways, we compared the fold of the NatA (PDB: 6HD5[36]) and NatB (PDB: 8BIP[37]) cryo-EM structures. We performed structural alignment of the subunits and scored their similarity with TM-align[38] (Fig. 1c). The catalytic subunits of NatA and NatB share a ~160 aa-long conserved structure with a high TM-score of 0.70, indicating the same fold. The similarity spans all domains, including four α-helices and seven β-strands, only excluding the intrinsically disordered regions (IDRs) unique to Naa10. The auxiliary subunits of NatA and NatB share a 554 aa-long conserved fold with a TM-score of 0.52, indicating they generally assume the same fold. The similarity spans most of the subunits, from their N-terminal, with only the last third of the subunits showing significant differences. However, they share an analogous domain organization with a ring-like structure of tetratricopeptide repeats (TPRs), completely wrapping the catalytic subunit. These results demonstrate that the catalytic subunits of NatA and NatB are highly conserved, in accordance with previous studies[16,39]. Meanwhile, the auxiliary subunits have diverged more but maintained the same overall fold. Thus, despite sharing high structural homology, each of the NATs subunits has a distinct yet opposite role in the co-translational assembly of their final complex.

Recent evidence suggests that even a single mutation can significantly impact proteins' interface formation and aggregation propensities, resulting in disease[40–42]. Hence, we hypothesize that co-translational assembly pathways may have evolved to rescue subunits accumulating destabilizing mutations. To decipher the biophysical properties governing thermostability, we performed all-atom molecular dynamics (MD) simulations[43], computing the dynamic properties of accessible conformational ensembles in an aqueous environment. We ran the MD simulations on the cryo-EM structures of NatA and NatB. To estimate individual amino acids' flexibility, we calculated each residue's root mean square fluctuation (RMSF) around their average position during MD (Fig. 1d–h; Supplementary Fig. 1b–i). Comparing even the most structurally conserved regions sharing the same fold (Supplementary Table 1), we detect a trend where co-translationally

engaged subunits fluctuate significantly more than their homologs (Fig. 1d). This trend is strong and persists at a wide range of temperatures (30 °C, 50 °C, and 100 °C; Fig. 1e–h). In all complexes, the subunits engaged as nascent chains (Naa15 and Naa20) exhibited

higher mobility when free but stabilized upon complexation. In contrast, their partner subunits (Naa10, Naa25) exhibited low mobility and high thermal stability, both in the complex and when free. The differences were accentuated at 100 °C, on the verge of denaturation, as

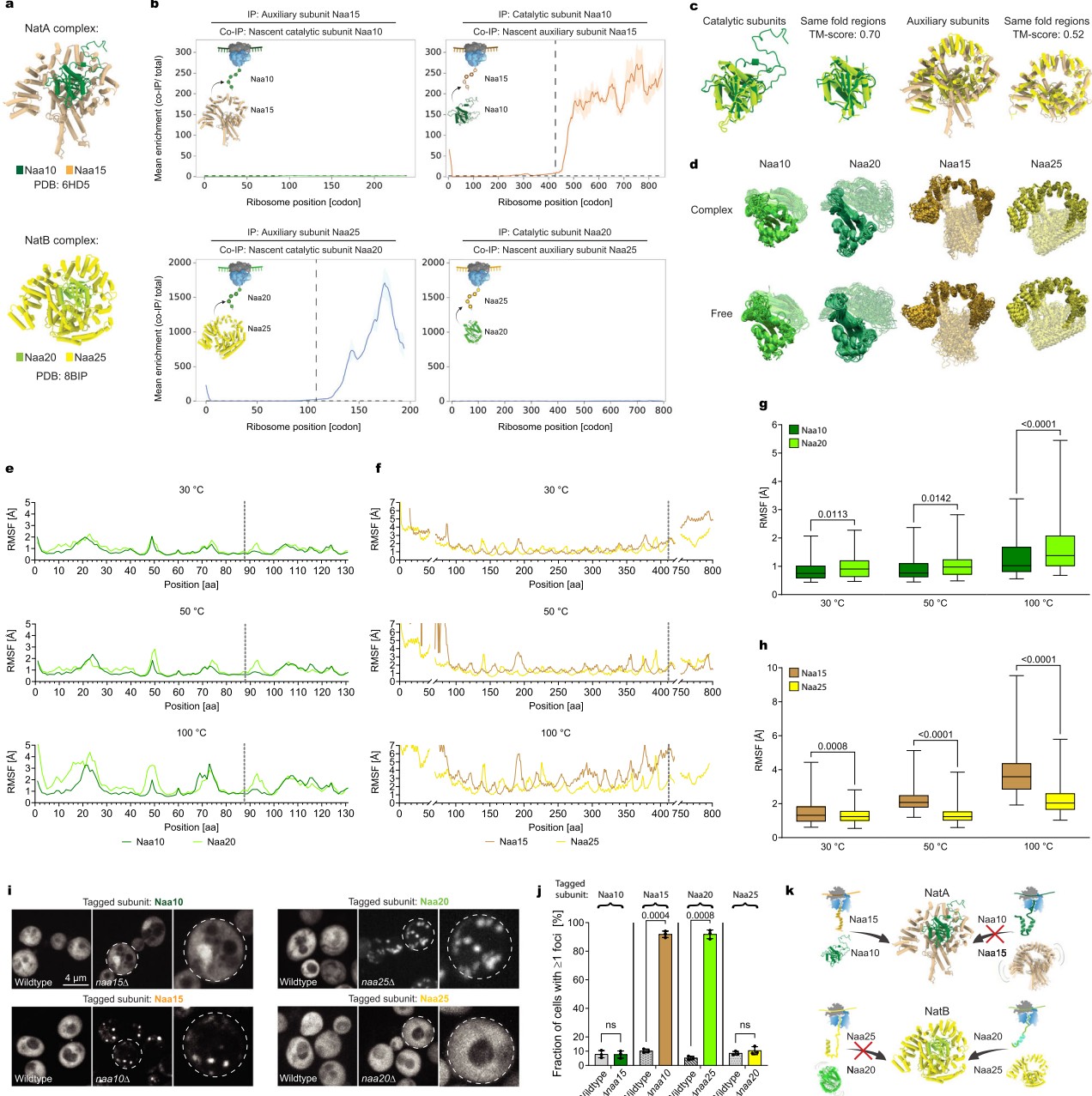

**Fig. 1 | NatA and NatB diverging co-translational assembly pathways correlate with opposite stability propensities. a** S. cerevisiae NatA and NatB heterodimers. **b** NATs subunits co-translational assembly analysis by SeRP, revealing divergent co-translational assembly pathways. Showing the mean with experimental variation between replicates shaded (*n* = 2 biologically independent experiments). The ribosome position at which the enrichment stably crosses the twofold threshold is indicated. **c** NatA and NatB structures superimposed, demonstrating their high similarity as determined by TM-Align. **d**–**h** Thermostability analysis of NatA and NatB by MD simulations. **d** Complex vs. free thermostability comparison of the catalytic and auxiliary subunits. MD simulation at 100 °C, at a 300 ns timeframe; 30 overlaid frames, taken at 10 ns intervals. Highlighted are the regions prior to co-translational interaction initiation (minimal regions) of Naa15, and Naa20 and its equivalent in Naa10, and Naa25. The long IDRs of Naa10 are not shown. **e**, **f** RMSF of the free catalytic subunits (**e**) and auxiliary subunits (**f**) during 300 ns of MD simulation at 30 °C, 50 °C, and 100 °C. At all temperatures. The dashed line

indicates the initiation of co-translational interaction. Shown are structurally conserved regions, as determined by TM-Align. **g**, **h** Catalytic subunits' (**g**) and auxiliary subunits' (**h**) RMSF boxplots of the structurally conserved regions, as in (**e**)–(**f**). Median is indicated with the box extending from the 25th to 75th percentiles and whiskers from minimum to maximum. Statistical analysis by unpaired two-sided t-test. *n* = 131 for Naa10, Naa20. *n* = 368 for Naa15, Naa25. **i** Solubility analysis of individual complex subunits tagged by GFP in vivo, in wildtype and in deletion strains expressing the tagged subunits orphaned from their partner subunit. A representative image is shown. **j** The fraction of cells displaying foci of GFP-tagged subunit. Data are mean ± SEM, *n* = 150 cells per sample from three biologically independent experiments, unpaired two-sided t-test (*p* value indicated). **k** Illustration of the assembly pathways of NatA and NatB, diverging in correlation with the subunits' opposite tendency for instability (grey lines), with the independently stable subunits (Naa10 and Naa25) engaging their nascent partner, thus facilitating their stability. Created with BioRender.com.

seen in their trajectories over time (Fig. 1d–h.; MD simulation Supplementary Movies 1, 2).

The catalytic subunits exhibit significant differences at the N-terminal region prior to the onset of co-translational assembly interactions. The differences in fluctuations decreased post this region (Supplementary Fig. 1h). Interestingly, the MD simulation (Supplementary Movie 1) revealed that the first ~50 amino acids of Naa20 undergo structural reorganization, with an alpha-helix briefly unfolding. The simulation of *Candida albicans* Naa20, a conserved homolog, displayed an even stronger behavior (Supplementary Movie 1; Supplementary Fig. 1d, h). This region is significantly stabilized in the presence of the partner subunit. The catalytic cleft also displayed much higher mobility in Naa20 compared to Naa10. Thus, we see diverging thermostability in highly conserved segments.

The RMSF of the auxiliary subunits reveals that Naa25 displays a significantly less flexible behavior than Naa15. The most prominent differences correspond to Naa15's N-terminal tendency to partially unfold, as well as the fluctuation of two long alpha helices involved in interface formation with the ribosome (Fig. 1f, h; Supplementary Fig. 1e, f, i; Supplementary Movie 2). This polarity in the behavior corresponds to our hypothesis that an anchor subunit, providing a scaffold for its partner to fold on, serves as a robust platform for the introduction of destabilizing mutations.

All RMSF calculations were done at thermodynamic equilibrium, as validated by the root mean square deviation (RMSD) of $C_\alpha$ atoms. We computed the RMSD for the complete protein (Supplementary Fig. 1j, k) and the TM-align defined core regions (Supplementary Fig. 1l, m) sharing a similar fold (Supplementary Table 1), measuring the deviation from the starting point. The RMSD graphs show that the subunits stabilize within less than 25 ns at all temperatures.

To test the thermostability predictions in vivo, aggregation assays of each subunit were performed in wildtype and in strains where the partner subunit was deleted. The "orphaned" condition impact on protein stability was then evaluated by imaging analysis of the protein's cellular distribution in live cells (Fig. 1i–j). The thermostability of the subunits predicted by MD simulations correlates with the aggregation propensity of the unassembled subunits. In both complexes, the two subunits engaged as nascent chains and which were predicted to be more flexible (Naa15 and Naa20) are also aggregation-prone, accumulating in foci when "orphaned" from their partner subunit. Meanwhile, their partner subunits remain stable, showing diffused staining both in wildtype and "orphaned" conditions. It is important to note that aggregation propensity is complex-specific, as deletion of the NatA catalytic subunit does not impact NatB catalytic subunit stability (Supplementary Fig. 1n). These results strengthen our hypothesis that subunit stability depends on co-translational complex assembly and is not caused by loss of function.

Interestingly, the homologous NatA and NatB complexes of two other yeast species (*Schizosaccharomyces pombe* and *Candida albicans*) display a similar asymmetric aggregation propensity of their subunits[44,45], suggesting a conserved co-translational compensatory mechanism across species. Our findings indicate that a particular aggregation-prone subunit is being engaged by its partner subunit co-translationally. Thus, co-translational interactions may provide a positive selection mechanism for driving functional divergence. Accordingly, the MD simulation suggests that NatB's catalytic cleft displays much higher flexibility than NatA, fitting with the difference in the complexes' substrate specificity. Where NatB catalyzes acetylation of much larger substrates of either Met-Asp, Met-Glu, Met-Asn, or Met-Gln, smaller substrates are catalyzed by NatA, including Ser, Ala, Gly, Cys, Thr, or Val after removal of the first Met by the protein N-terminal methionine excision pathway[15,16]. Thus, higher flexibility can be compensated by complexation, already during translation (Fig. 1k).

## Energy profiles govern co-translational complexation

To investigate the energy landscape of the diverging complexation of NatA and NatB, we first employed pyDock bindEy module[46]. pyDock predicts a total complexation energy of −211 kcal/mol for NatA compared to only −118 kcal/mol for NatB (Supplementary Table 4). The catalytic subunits share the same core fold, differing only in two intrinsically disordered regions (IDRs) unique to NatA. To test the impact of these diverging regions, we next computed the pyDock energy for NatA without these IDRs, by removing residues 50-90 and 196-236 of NatA (Supplementary Table 4). It resulted in almost a twofold decrease in complexation energy, to a total of −106 kcal/mol, similar to NatB's. This decrease suggests that the IDRs of NatA are crucial for the formation and stabilization of the complex. Accordingly, anchor residues within IDRs have been recently suggested to direct folding and binding to interaction partners[47,48].

To identify anchor residues that are energetically crucial to interface formation, we utilized the molecular mechanics Poisson−Boltzmann surface area (MMPBSA)[49] method. MMPBSA computes the binding free energy of complexation contributed by each residue, averaged over an ensemble of representative conformations. This method accurately captures the interface formation dynamics, compared to only a single conformation captured by the crystal structure. We computed MMPBSA interaction energy values per residue for 500 conformations from the last 20 ns of the 300 ns long production simulations (Fig. 2a, b). The results show that the energy distribution diverges between co-translationally engaged subunits and their stable partners. In the NatA auxiliary subunit (Naa15), most interface energy contribution is clustered before the onset point, establishing early interface formation (residues ~255-355; Fig. 2a). Notably, 13 of the 15 Naa15 highest energy residues, the interface "hotspots", are located in the minimal region, at the interface with the long IDRs of the catalytic subunit. Meanwhile, hotspots are more evenly distributed along the stable catalytic subunit (Naa10), acting as a scaffold to its nascent partner (Fig. 2a). In NatB, the auxiliary subunit's (Naa25) hotspots are symmetrically distributed, whereas in the co-translationally engaged catalytic subunit (Naa20), they are concentrated in a cluster before the onset point, contributing most of the interaction energy before assembly initiates (Fig. 2a). Thus, both co-translationally engaged subunits display the same pattern of hotspots clustering before the onset of co-translational assembly interactions. In contrast, their stable partners display a symmetric distribution from the N- to the C-terminal. Analyzing the cumulative ΔΔG along the subunits, we can see a gradual increase in interface energy for the stable partners. Their co-translationally engaged subunits, on the other hand, display a sharp rise of ~35 amino acids before co-translational interaction initiates, as we detect a dense clustering of interface hotspots (Fig. 2b; the distributions were found to be significantly different based on two-sample Kolmogorov-Smirnov test). The ~35 amino acids gap correlates with the nascent-chain length that can be accommodated in the ribosomal exit tunnel, according to cryo-EM studies[50]. Thus, the onset of co-translational assembly interactions correlates sharply with the exposure of the interface hotspots cluster at the ribosome exit tunnel.

The asymmetric distribution of hotspots detected by the MMBPSA conformations ensemble is in stark contrast to the interaction residues we detect in the single conformation captured in the crystal structure (Fig. 2c, d). In the crystal, interaction residues (determined by distance <4 Å between the subunits $C_\alpha$ atoms) are symmetrically distributed in all subunits, in NatA and Nat B.

Together, these results suggest energy profiles can provide profound insight into the nature of complexation and help predict the formation of meta-stable interfaces and the onset of co-translational subunits' engagement. The energy profiles and co-translational analysis by SeRP suggest a different putative mechanism for NatA and NatB assembly. In NatB, the results point at a single-step mode of

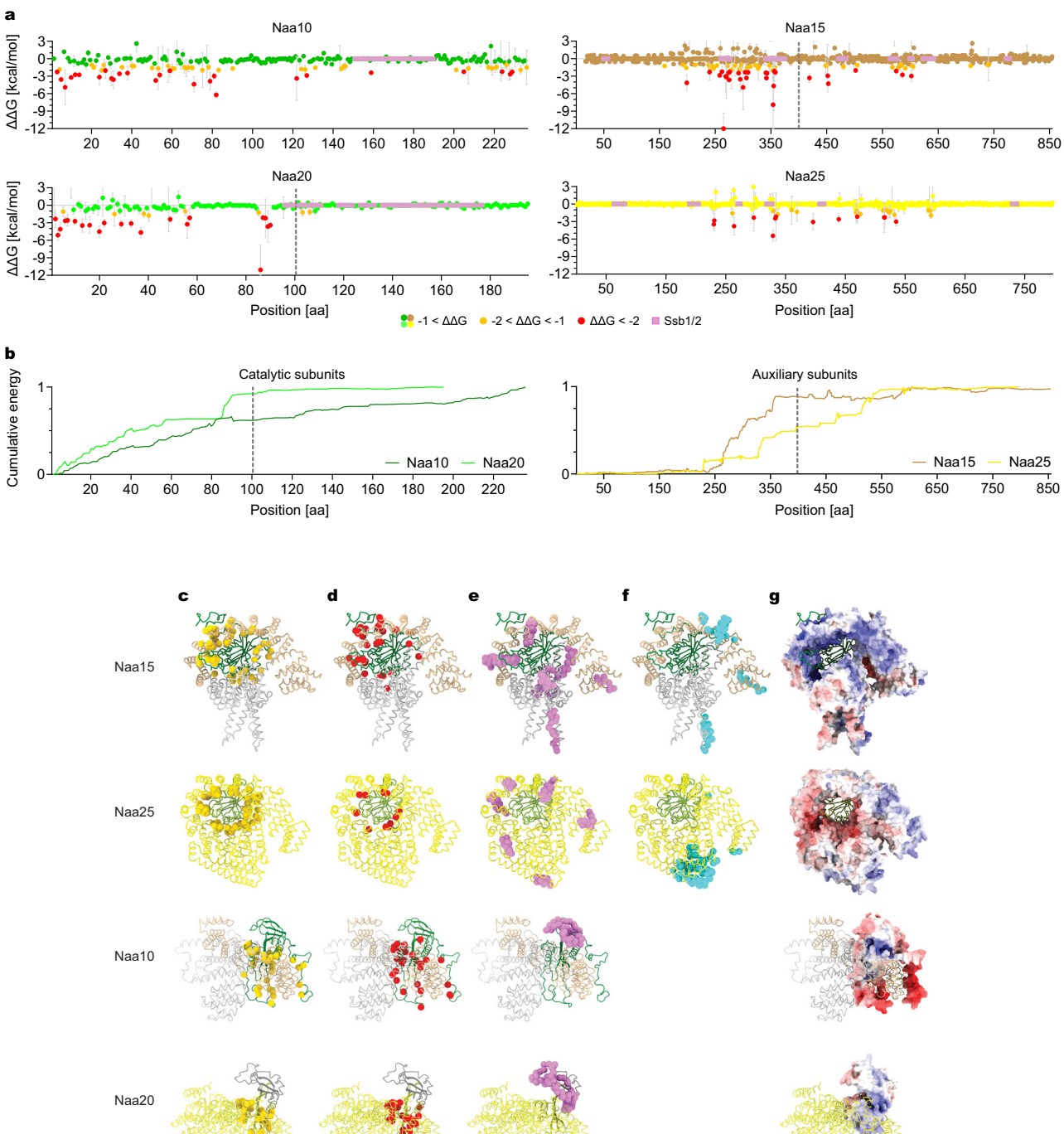

**Fig. 2 | Energy profiles govern co-translational interface formation. a** Interface energy contribution (ΔΔG [kcal/mol]) per residue in each subunit, as calculated by MMPBSA along the last 20 ns of the 300 ns simulations, at 25 °C. 500 evenly spaced frames were sampled. Both NatA auxiliary and NatB catalytic subunits have a cluster of residues crucial for interface formation just before co-translational interaction initiates, indicated by a dashed line. Mean ± SD, $n = 3$ independent production simulations. **b** Cumulative binding free energy, with all the contributions normalized to $\Delta G < 1$. The cluster of residues crucial for interface formation can be seen as the steep fall in the cumulative energy, 30 residues before the co-translational interaction initiates, indicated by a dashed line. The cumulative distributions were found to be significantly different based on two-sample Kolmogorov-Smirnov test with a $p$ value = 1.688e-09 for the catalytic subunits and $p$ value = 2.313e-07 for the auxiliary subunits. **c** Interface residues according to the cryo-EM structures, as determined by $C_\alpha$-atoms in a 4 Å proximity from a $C_\alpha$-atom of its partner subunit, displayed as orange spheres. **d** Interface $C_\alpha$-atoms which contribute $< -2$ kcal/mol ΔΔG to interface formation, displayed as red spheres. **e** Chaperones binding sites, as obtained by SeRP (data derived from accession code GSE93830), displayed as violet spheres. **f** Ribosome binding sites, as obtained from solved structures of NatA (PDB: 6HD7) and NatB (PDB: 8BIP) on the ribosome, displayed as light blue spheres. $C_\alpha$-atoms within 4 Å from any heavy atoms of the ribosome are marked. **g** Electrostatic potentials as calculated by the Adaptive Poisson–Boltzmann Solver[81,82] (APBS). The colors [red, white, blue] are mapped on a [−10, 0, 10] range in units of kJ/mol/e. NatA and NatB display diverging electrostatic potentials, contributing to the diverging interface energy profiles. NatA catalytic subunit's unique IDRs, participating in interface formation, display strong negative potential, which is complemented by the NatA auxiliary subunits' positive potential.

assembly: The catalytic subunit is symmetrically bound on all its perimeter by the auxiliary subunit immediately upon exposure of the catalytic's minimal region out of the ribosome exit tunnel. Thus, the thermostable auxiliary subunit can serve as a scaffold, stabilizing the more flexible catalytic subunit already during synthesis. In contrast, the results suggest NatA's complexation is a two-step mechanism. First, the more stable catalytic subunit binds the minimal region of the auxiliary subunit. The highly charged IDRs of NatA (Fig. 2g) provide a meta-stable partial interface, as most of the interface binding energy contribution is established during this first step. Next, the auxiliary subunit synthesis proceeds, and as the interface emerges from the ribosome exit tunnel, it continues to wrap around its partner, which provides a stable platform. Notably, all known NatA complexes, from yeast to humans, show long stretches of putative IDRs[14]. These unique IDRs are not present in NatBs, suggesting a conserved function in forming strong interfaces during assembly.

## Ribosome-associated chaperone Ssb1/2 guards assembly hotspots

We next analyzed assembly interplay with the ribosome-associated Hsp70 canonical chaperone Stress-Seventy subfamily B (Ssb1/2). Ssb1/2 action was shown to be coordinated with assembly, transiently binding partially synthesized interface domains, protecting them from misfolding[6,51,52]. SeRP analysis of Ssb1/2 co-translational association with the NatA and NatB subunits (data derived from accession code GSE93830[51]) indicates it transiently engages several clusters of hotspots in both of their auxiliary subunits, protecting them prior to assembly (Fig. 2d, e). For example, residues R354 and R355 of Naa15, contributing the most energy to interface formation in the subunit (around −8 kcal/mol each), are protected by Ssb1/2 binding just before the onset of complex assembly interactions. This correlates well with the high abundance of arginine residues in the consensus motif of Ssb1/2[51]. Furthermore, Ssb1/2 co-translational binding was correlated to hyperflexible ribosome binding regions identified by MD thermostability analysis (Fig. 2f). These results suggest interface residues characterized by low thermostability are guarded by ribosome-associated chaperone Ssb1/2, prior to co-translational assembly interactions.

In NatA and NatB catalytic subunits, the Ssb1/2 binding sites protect mainly the C-term region, which forms the upper part of the substrate binding sites (Fig. 2e). The catalytic subunits are much shorter than the auxiliary subunits and harbor a cluster of hotspots spanning their first ~50 aa (Fig. 2a, d). Ssb1 and Ssb2 were shown to be depleted from such extreme N-terminal domains[51], as they require longer polypeptide-chain exposure out of the ribosome exit tunnel for interaction. Still, we observed a similar trend to the auxiliary subunits, where the smaller, thermolabile subunit Naa20 displayed nearly twice as many binding residues to Ssb1/2 compared to Naa10 (Fig. 2e, Supplementary Table 2).

The first ~50 aa may fold independently or rely on the action of other co-translational chaperones. Thus, we next analyzed the T-complex protein Ring Complex (TRiC) chaperone co-translational binding to the NATs subunits[52]. TRiC binding was not detected for NatB subunits. For NatA subunits, binding of TRiC was also hardly detected, interacting only with a very short segment in the catalytic subunit (residues 103–109; Supplementary Fig. 2; Supplementary Table 2). Extreme N-terminal domains were suggested to fold independently, possibly even initiating folding inside the ribosome exit tunnel[53,54]. We hypothesize that the N-terminal domains of the catalytic subunits may fold independently yet require fast stabilization. Accordingly, we detect early co-translational assembly interactions, which can stabilize NatB catalytic subunit N-terminal hotspots cluster exposure.

## Interface hotspots are critical for co-dependent subunits' stability, complex formation, and function

To study the impact of hotspot residues on the system, we utilized MD simulations to predict complexation behavior following mutations

(Fig. 3a, b; Supplementary Movie 3). In the NatA system, residues R354 and R355 of the auxiliary subunit contribute the highest energy to interface formation, while also contributing to Ssb1/2 binding. Based on the MMPBSA energy decomposed by residue, R354 and R355 strongly interact with several glutamic and aspartic amino acids of its catalytic partner, mainly D53, E55, and D56 (Fig. 3a, Supplementary Table 6). Thus, R354 and R355 form a highly energetic interface with the large, and strongly negatively charged, surface on the first large IDR loop of the catalytic subunit (Fig. 2g). We hypothesized that replacing these arginine residues with glutamates would disrupt these interactions, creating an electrostatic repulsion. To test our hypothesis, we simulated the complex dynamics of the wildtype system, R354 and R355 mutant to alanines, and R354 and R355 to glutamates mutant, for 300 ns at 25 °C (Fig. 3b). Then we computed the pyDock bindEy energy for all systems. In the R354A, R355A mutant system, the total interaction energy remains very close to the wildtype value (−203 kcal/mol vs. −211 kcal /mol; see Supplementary Table 4), as MD suggests interaction can still occur over the system evolvement, although is less stable. However, in the R354E, R355E mutant, the complexation energy computed by pyDock is much lower (−178 kcal/mol), as MD suggests the mutations to glutamic acid introduced a strong repulsion in the area, leading to loss of local structure and interface interactions, as shown in Fig. 3b.

To investigate the impact of hotspot mutations in vivo on co-translational assembly interactions we generated yeast strains expressing mutated variants in *NAA15* endogenous locus by CRISPR/Cas9[55]: R354A, R355A, or R354E, R355E. Its catalytic partner, *NAA10* was C-terminally tagged with HA[56] in these strains to allow for analysis of co-translational interactions in a quantitative manner by RIP-qPCR[10,57] (Fig. 3c). We immunoprecipitated the tagged catalytic subunits (Supplementary Fig. 3a) and analyzed their association with the auxiliary partner's mRNAs. R354E, R355E double mutant significantly reduced co-translational assembly interactions by more than 10-fold compared to the wildtype, while R354A, R355A double mutant led to 3-fold reduction (Fig. 3d). EDTA treatment, leading to ribosomal dissociation[8,12,30,58], confirmed that these protein-RNA interactions depend on polysome integrity, thereby supporting that they reflect co-translational assembly events between the immunoprecipitated protein (Naa10) and its nascent partner (Naa15), emerging from the ribosome. Thus, the mutation of interface hotspots significantly impacts co-translational assembly interactions, in line with the interface disruption predicted by MD simulations.

To explore the mutants' impact on aggregation propensity, we next introduced these mutations to fluorescently tagged *S. cerevisiae* (mCherry-Naa15) in its endogenous locus by CRISPR/Cas9[57] (Fig. 3e, f). Naa15 aggregation propensity was analyzed by imaging and compared to wildtype, serving as a solubility control, and a strain deleted of its partner subunit in the complex (*naa10Δ*), exhibiting strong aggregation (Fig. 3e[6]). In accordance with the MD predictions and RIP-qPCR results the R354E, R355E mutant displayed strong aggregation, even under physiological conditions (30 °C). Quantification of the average aggregates per cell (Fig. 3f) demonstrated that R354E, R355E mutant aggregation phenotype was as significant as deletion of the entire partner subunit (*naa10Δ*). Heat-shock stress, mild and severe (37 °C for 1 hour or 42 °C heat-shock for 10 min), showed similar results (Fig. 3f). Inhibition of protein synthesis by cycloheximide just prior to severe heat-shock strongly inhibited foci formation, suggesting Naa15 is most sensitive to misfolding during synthesis, and thus might depend on co-translational assembly interactions to achieve its native fold. Furthermore, R354E, R355E may also disrupt association with Hsp70 chaperones, such as ribosome-associated Ssb1/2 as Ssb1/2 substrates are significantly disenriched for EE-containing motifs[51]. R354A, R355A mutations did not cause significant aggregation at 30 °C. Upon heat-shock, aggregates appeared, increasing in number with the severity of the heat-shock (Fig. 3e, f). MD simulations suggest R354A, R355A

mutants allow for interface formation, although with lower stability, due to lower interface energy, as confirmed by RIP-qPCR. Foci elevated levels with heat stress suggest interface stability is indeed reduced, compared to wildtype. We also analyzed the impact of hotspot

disruption in the catalytic subunit. Each of Naa10's hotspots displayed a much smaller interface energy contribution. Thus, a triple mutant was generated: Naa10 P2A, I5A, R7A. Naa10 hotspots mutation impact on mCherry-Naa15 was exhibited by elevated levels of foci formation

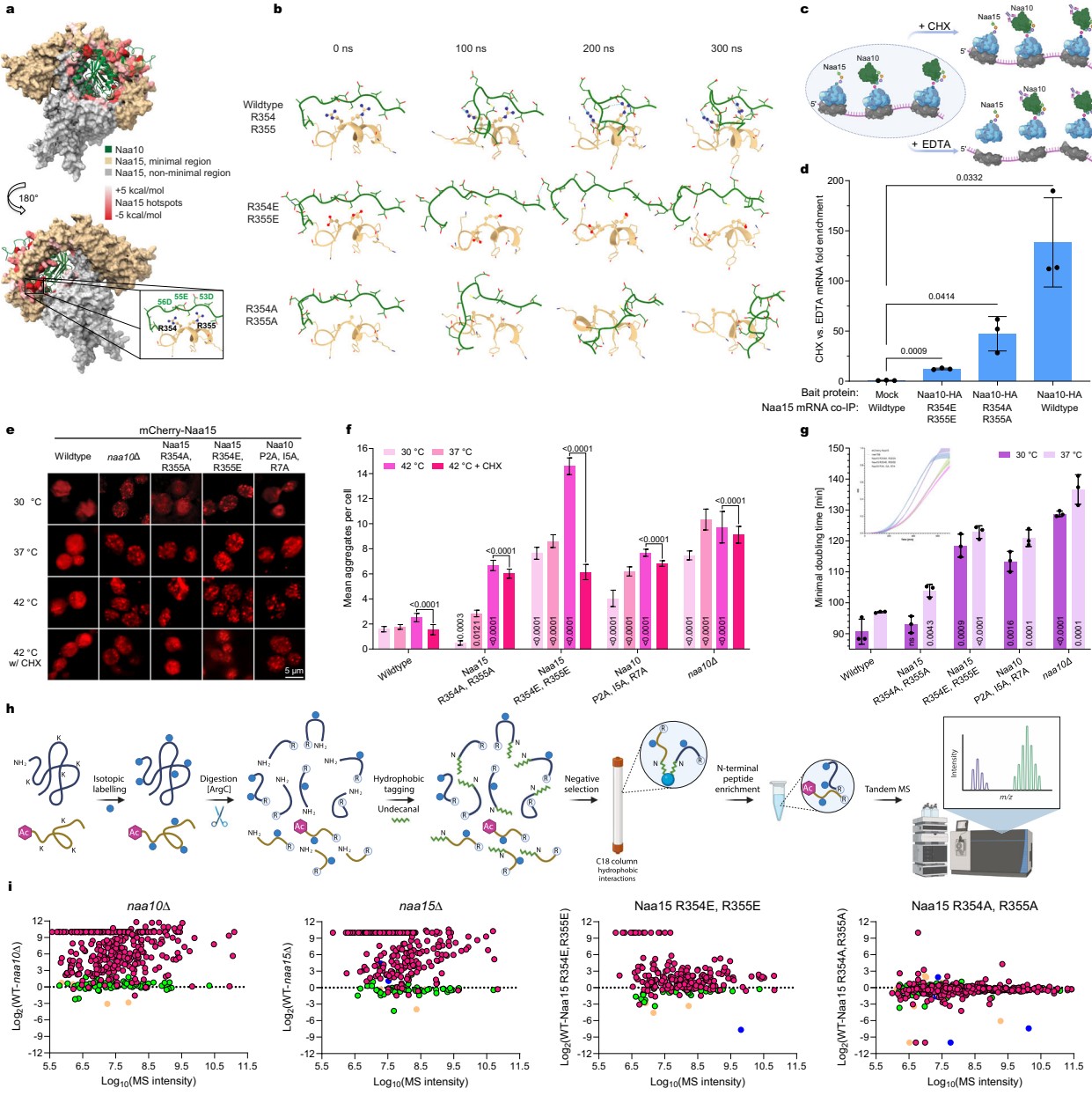

**Fig. 3 | Interface hotspots are critical for co-translationally engaged subunits' assembly, stability, and function. a, b** Impact of residues' mutation on interface formation of NatA, as simulated by MD. **a** Interface hotspots overlaid in surface presentation, graded from white ( + 5 [kcal/mol]) to red (−5 [kcal/mol]) by their ΔΔG contribution to interface formation. Zoom-in on Naa15 R354, R355, the residues displaying the highest contribution to NatA interface energy. **b** MD simulations of wildtype and the indicated mutatnts' impact on the interface. Zoom in on R354, R355 area with hydrogen bonds indicated by dashed lines. Simulated over 300 ns at 25 °C, n = 3. **c, d** Interface hotspots mutation impairs co-translational assembly. **c** RIP-qPCR experimental scheme, quantitively assessing Naa15 mutants' impact on co-translational interaction levels. Created with BioRender.com. **d** The indicated Naa15 mRNAs strains association with HA-tagged Naa10 (bait), comparing cell lysates treated with cycloheximide vs. EDTA (mean ± SD, n = 3 biologically independent experiments, unpaired two-sided t-test). All mutants significantly impair co-translational assembly interactions. **e, f** Mutation analysis of interface hotspots reveals their dramatic impact on proteins' misfolding propensity. **e** Representative

images of wildtype and indicated mutated strains, exhibiting the impact of hotspot mutation on mCherry-tagged Naa15 solubility. **f** Quantification of mean aggregates per cell, utilizing Imaris (mean ± SEM, n = 150 cells per sample from three biologically independent experiments, unpaired two-sided t-test against the wildtype under the same conditions, except 42 °C with cycloheximide (CHX) samples which were tested against 42 °C of the same strain). **g** Mutation analysis of interface hotspots reveals their significant impact on growth. The minimal doubling time of wildtype and the mutated strains, extrapolated from growth curves (see Supplementary Fig. 3b corresponding to the small graph at the top left; mean ± SD, n = 3 biologically independent experiments, unpaired two-sided t-test against wildtype under the same conditions). **h, i** Interface hotspots mutation impairs NatA function. **h** N-terminomics experimental scheme utilizing hydrophobic tagging-assisted N-termini enrichment (HYTANE), followed by LC-MS/MS analysis. Created with BioRender.com. **i** Global N-terminome acetylation abundance ratios of Nt-acetylated peptides in the indicated mutants relative to the wildtype, plotted against their mass spectrometry (MS/MS) intensity – heavy plus light channels.

compared to wildtype, which increased with heat stress severity. This corresponds to the energy of the mutated hotspots and the predicted change in interface caused by the mutations.

Growth assays were conducted for all generated strains in optimal conditions (YPD, 30 °C) and under mild heat stress for 24 hours (YPD, 37 °C; Fig. 3g)[58]. The R354A, R355A mutant behaved as the wildtype, while the R354E, R355E mutant exhibited a severe growth phenotype, closely fitting the deletion strain. The triple mutant of the catalytic subunit (Naa10 P2A, I5A, R7A) also inhibited growth, though less severe than the Naa15 R354E, R355E mutant. These results demonstrate that impairing NATs' interface association leads to severe aggregation and inhibits growth.

To assess the impact of co-translational assembly impairment on the NATs' function, we compared the relative abundance of N-terminally-acetylated peptides of the wildtype versus *naa10Δ*, *naa15Δ*, and the two *NAA15* mutants: R354A, R355A and R354E, R355E. The N-terminal peptides were isotopically labeled and enriched using the N-terminomics analysis method HYTANE[58] (hydrophobic tagging-assisted N-termini enrichment), followed by liquid chromatography with tandem mass spectrometry (LC-MS/MS) analysis, to achieve global N-terminome results[58–60] (Fig. 3h). In Fig. 3i, abundance ratios of the mutant relative to the wildtype of Nt-acetylated peptides are plotted against their MS intensity, which was used to estimate their abundance in the cell. NatA substrates were affected by the mutations with different severity, while NatB and NatC substrates were not affected and centered around zero. *naa10Δ* displayed the most drastic change compared to the wildtype, with NatA substrates showing a sharp decline in their acetylation levels. R354A, R355A mutant maintained highly similar levels of N-terminal acetylation by NatA as the wildtype. However, R354E, R355E mutant exhibited a strong decrease in the acetylation of its substrates, which remain free (Extended Data Table 2). Thus, the results demonstrate that co-translational assembly interaction impairment significantly affects the complex's function. Taken together with the RIP-qPCR, imaging, and the growth rates results, the in vivo results strongly suggest that hotspots' mutations can lead to a sharp decline in co-translational assembly interactions, which in turn impair complex assembly, stability, and function.

Conservation analysis revealed that missense mutations in human NATs, causing neurodevelopmental and neurodegenerative diseases, can disrupt putative hotspot clusters (Supplementary Fig. 5). For example, in the catalytic subunit of human NatA, D10G and L11R (D12 and I13 in *S. cerevisiae*) have been linked to several neurodevelopmental pathologies[61]. These missense variants are far removed from the catalytic site; however, they caused a significant reduction in the observed catalytic activity. Simulating the mutated proteins compared to the wildtype human Naa10 revealed their impact on its fold, causing a shift in the conformation of two α-helices at the N-terminal harboring most of the interface energy (Supplementary Fig. 5). Thus, mutations can impair the ability to form the interface with the auxiliary subunit, thus inhibiting the complex's substrate binding and catalytic activity. Similarly, the conserved Naa10 S37 (S39 in *S. cerevisiae*) mutation to proline was found to cause Ogden syndrome, impairing complex assembly and catalysis[24]. The alpha-helix harboring S37 encompasses six hotspots, to a total energy of −29.4 kcal/mol (Supplementary Fig. 5b, Supplementary Table 9), showing S37P destabilizes the alpha helix, disrupting assembly and catalysis. Several other missense variants in conserved residues, such as Naa15 K338D[62,63] (K358 in *S. cerevisiae*), all involved in developmental delay and microcephaly, can cause similar defects in assembly, as they disrupt predicted hotspots clusters.

Overall, there is a strong agreement between our results and the predictions MD simulations provided. These results demonstrate the predictive power of MD combined with MMPBSA energy profiles for co-translational assembly interactions.

## A co-translational assembly energy profile trend emerges

In this study, we aimed to assess the predictive capability of energy profiles concerning the onset of co-translational interactions in specific protein complexes. Our investigation expanded to include the phosphofructokinase complex (PFK), anthranilate synthase complex (TRP), and nucleoporin Nup85-Seh1 subcomplex in yeast. Prior studies using SeRP revealed the onset of co-translational interactions for these complexes[6,8], so we could correlate directionality and onset with energy profiles (Fig. 4a). Due to crucial segments missing in the solved structures, we employed AlphaFold-Multimer[31] to model the complete complexes, on which we ran MD simulations followed by MMPBSA analysis. The robustness of this method was demonstrated for NatB, comparing its cryo-EM derived analysis to one based on AlphaFold-Multimer modeling (Supplementary Fig. 4).

The PFK complex is composed of alpha (Pfk1) and beta (Pfk2) subunits with 50% sequence identity and a TM-score of 0.79 (Fig. 4b, c). Despite sharing high structural similarity, these subunits exhibit distinct onset of co-translational interactions. For Pfk1 onset occurs when the first ~200 residues of Pfk1 nascent-chain are exposed. In Pfk2, onset occurs much later, when ~450 amino acids are synthesized, and enrichment levels fluctuate much more compared to Pfk1, until the end of synthesis[6]. Notably, the energy profiles closely correlate with these onset points. Pfk1 displayed a prominent clustering of hotspots in its first 200 residues, whereas Pfk2 demonstrated a more even distribution of hotspots along its first 450 residues, and overall along the ORF (Fig. 4a). The differences in the energy profiles led us to investigate the subtle structural differences in the minimal regions, synthesized prior to co-translational assembly onset (Fig. 4b, c). Pfk2 has a distinct 50-amino acid IDR (aa 146-198) that serves as a long linker between two domains that display high similarity in both subunits: the N-terminal glyoxalase-like domain and the middle phosphofructokinase domain. This extended IDR allows the middle phosphofructokinase domain to adopt a 180° flipped position relative to its N-terminal domain (Fig. 4c). This enables an asymmetric interface formation leading to an asymmetric co-translational assembly pathway. Thus, similar to the NATs case, energy profiling allows for onset prediction, where structural features are too homologous for distinction.

The TRP heterodimer consists of the highly conserved Trp2 and Trp3 subunits. Energy profiling exhibited a robust correlation with the onset of interactions in Trp3, with significant hotspot clustering observed just prior to the onset (Fig. 4a). However, for Trp2, despite detecting hotspot clustering around 350 residues, no co-translational interactions were observed. This discrepancy can arise from the intricate fold of Trp2. Its interface can only form after the synthesis of its C terminus as adjacent β-strands are separated by a ~150 aa gap (β-strand in position 299-303 aa is connected to β-strand in position 445-449 aa, for example; Fig. 4d). Thus, co-translational interface formation cannot occur, as segments that are synthesized far apart are co-dependent on each other for folding and stability.

Regarding the nucleoporin subcomplex, energy profiling of Seh1 demonstrated an evenly-distributed interface energy along the ORF, encompassing its extreme C terminus. This distribution coincided with the lack of interactions detected during the synthesis of this subunit. In contrast, Nup85 displayed a strong clustering of hotspots in its extreme N terminus, contributing over 80% of its interface interaction energy. Nevertheless, co-translational complexation was observed only after the synthesis of the second hotspot cluster. The first cluster region is highly flexible. We hypothesize that the first cluster (aa 47-95) can only stably fold upon the synthesis of the second cluster (approximately aa 450-550), as they are closely interacting, including several hydrogen bonds, despite their distance in the linear sequence (Fig. 4e). Thus, like Trp2, only after the synthesis of the second cluster can the entire interface form. In contrast to Trp2, this second cluster is exposed at the ribosome exit tunnel at aa ~565, when approximately

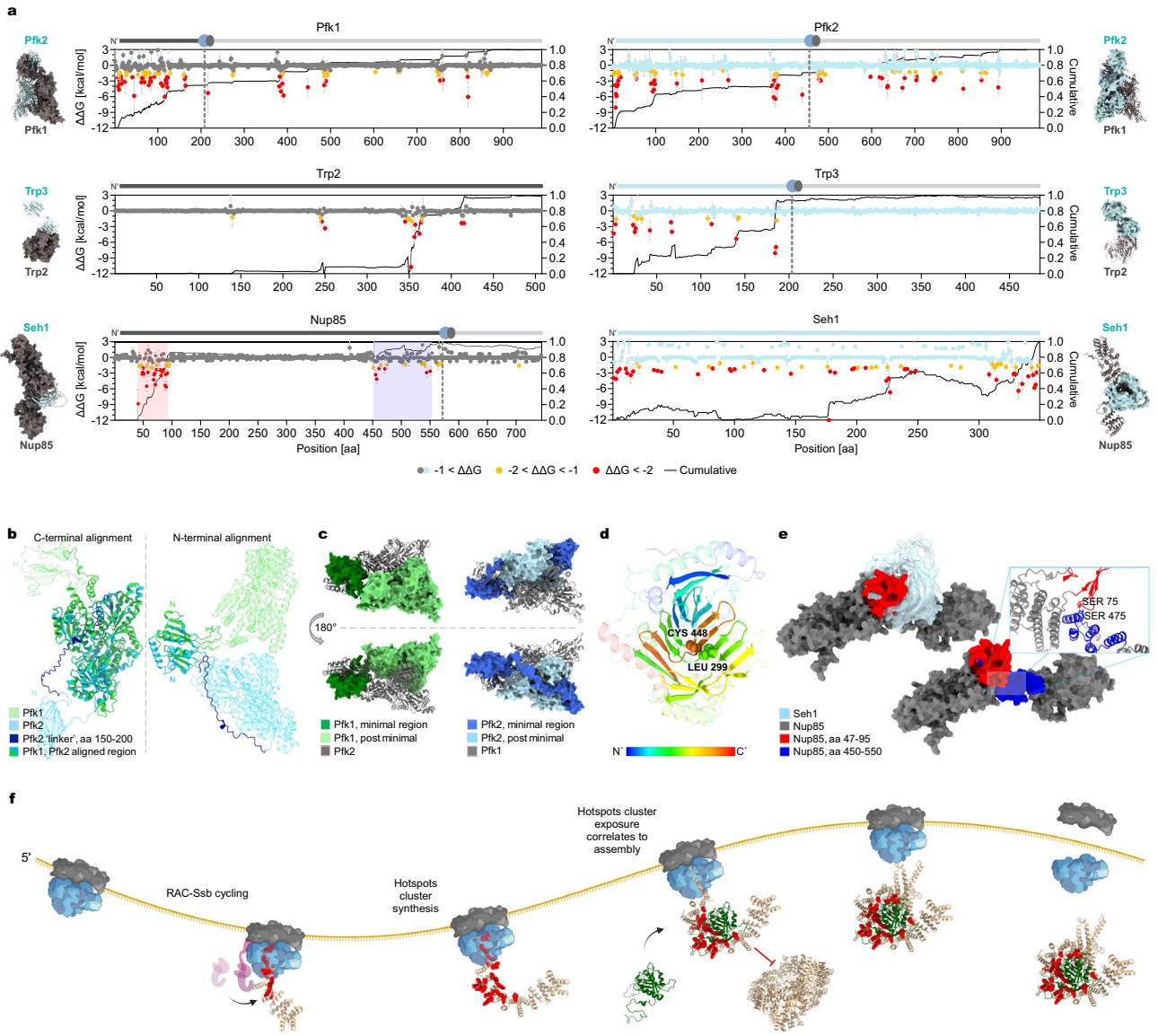

**Fig. 4 | Complexes which assemble co-translationally display interface energy clustering. a** Interface energy contribution (ΔΔG [kcal/mol]) per residue in indicated subunits of TRP, PFK, and Nup85-Seh1 nuclear pore subcomplex. The complexes were modeled by AlphaFold-Multimer. ΔΔG calculated by MMPBSA as in Fig. 2a. Cumulative binding free energy is indicated by a black continuous line for each subunit. Co-translational assembly onset is indicated by a vertical dashed line, where the nascent chain is engaged by its partner (data extrapolated from previous studies[6,8]). Nup85's two hotspot clusters highlighted in red and blue correspond to Fig. 4e. Mean ± SD, n = 3 independent production simulations. **b** Comparison of Pfk1 and Pfk2 demonstrates their high structural similarity, except for a unique 50 aa IDR (dark blue) in Pfk2, as determined by TM-Align. Minimal regions, prior to co-translational complex assembly (in a darker shade) versus the post-minimal regions. Left: Aligned aa 1-150 with RMSD of 0.8 Å between 62 pruned atom pairs. Right: Aligned aa 150-959 aa with RMSD of 0.5 Å between 707 pruned atom pairs. **c** Pfk2's unique IDR allows its N-terminal domain to adopt a distinct position compared to Pfk1 in the complex. **d** AlphaFold-generated model of Trp2, rainbow-colored from N- to C-terminal, displaying its intricate fold, allowing interface domains to fold only after C-terminal segments synthesis. **e** AlphaFold-Multimer predicted structures of NPC subcomplex Nup85-Seh1 (left) and Nup85 alone (right). Nup85's two hotspot clusters are highlighted in red (aa 47-95) and blue (aa 450-550), surface display. Zoom-in: intramolecular hydrogen bonds between the first and second hotspot clusters suggests their stabilization is co-dependent, only allowing for co-translational complex assembly interactions after the second cluster's synthesis. **f** Co-translational complex assembly suggested mechanism: hotspots cluster partial exposure at the ribosome exit tunnel is protected by chaperones' holdase activity. The onset of co-translational complex assembly interactions occurs immediately upon hotspot cluster complete synthesis and exposure. Onset may be delayed due to hotspot cluster regions stabilization by downstream structural elements synthesis. Impairment of co-translational assembly interactions leads to the misfolding of subunits harboring hotspot clusters, conferring high flexibility and consequently partner-dependent stability. Created with BioRender.com.

200 aa of Nup85 are yet to be synthesized, allowing for co-translational assembly interactions.

In summary, our findings underscore the predictive power of energy profiling in identifying co-translational interactions and the onset during translation. A significant fraction of the interface energy is contributed by only a few amino acids, which are clustered and synthesized just before the onset of assembly interactions. Furthermore, the vectorial order of interface synthesis plays a critical role in guiding the initiation of co-translational assembly interactions. Hotspots scattered along the linear sequence can form clusters in the 3D fold. Hence, our results suggest a combination of energy profiling and analysis of hotspots 3D clustering are strong predictors of co-

translational assembly interactions. These insights enhance our understanding of the dynamic molecular processes governing protein complex formation during translation (Fig. 4f).

## Discussion

In this study, we conducted an extensive investigation into the assembly pathways of NatA and NatB. Combining selective ribosome profiling with exhaustive molecular dynamics, we have shown that the diverging assembly trajectories for these two homologous heterodimers are directed by distinct interface energy distributions. Specifically, we found that the NatA catalytic subunit engages with its unstable auxiliary counterpart in a co-translational fashion, effectively preventing misfolding. Conversely, the NatB complex follows an opposite route, wherein its catalytic subunit is co-translationally engaged by its auxiliary partner subunit. Consequently, the NatB catalytic subunit is more prone to misfolding, leading to aggregation when isolated from its auxiliary partner. Notably, despite their structural and sequence homology, both NatA and NatB catalytic subunits adopt differing assembly pathways, raising questions regarding the origin of these contrasting co-translational assembly strategies.

To address this query, we employed a combination of in vivo methodologies to capture co-translational interactions, analyze complex functionality, and evaluate subunit stability. Furthermore, we harnessed several in-silico tools, including MD simulations, modeling, and pyDock energy predictions. By integrating these complementary approaches, we gained deeper insights into the key energetic, flexibility, and thermostability characteristics that govern the divergent assembly pathways of NatA and NatB complexes. Particularly, our MD simulations uncovered distinctive thermostability profiles and interface hotspot distributions. Co-translationally engaged subunits, such as NatA auxiliary and NatB catalytic subunits, exhibited an uneven distribution of interface "hotspots," with these hotspots clustering around 35 amino acids before the onset of co-translational assembly interactions. In contrast, subunits that interact post-translationally demonstrated a more symmetrical and even distribution of interface energy along the entire ORF. These latter subunits exhibited higher thermodynamic stability, reduced flexibility in conserved regions, and fewer chaperone binding sites. IDRs also contributed significantly to establishing co-translational interfaces, providing solubility and stability independent of the folding process.

The asymmetrical energy distribution, together with co-translational interactions and diverging thermostability, strongly indicates that the interface energy distribution within each subunit governs co-translational assembly interactions. Remarkably, the data suggest that a small number of key hotspots, spatially clustered within a subunit's linear or three-dimensional structure, play a central role in initiating co-translational interactions, effectively anchoring interface formation during synthesis. Furthermore, the energy profiles propose distinct hypothetical mechanisms for the complexation of NatA and NatB. In the case of NatB, a single-step mode of assembly is proposed, where the catalytic subunit exposes its lower interface region on the ribosome, promptly engaging with the auxiliary domain to confer stabilization. Conversely, NatA complexation unfolds in two steps. Initially, the minimally folded region of the auxiliary subunit binds its partner, though not yet completely enveloping it. The majority of interface binding energy is established in this initial step, facilitating early interface formation. Subsequently, gradual development of the remaining interface between subunits takes place, safeguarded against misfolding by Ssb1/2 binding sites.

Subunits that present a cluster of high-energy residues, inducing instability when they are orphaned from their partner subunit, can rely on co-translational assembly interactions to guide proper folding and assembly pathways. Hence, it can be speculated that the NatB catalytic subunit, which is responsible for acetylating larger substrates compared to NatA[14,16], might have evolved to depend on co-translational interactions to rescue its folding process. This reliance could have facilitated the emergence of a more flexible catalytic cleft in NatB, thus expanding its substrate range. This hypothesis is substantiated by MD simulations of the catalytic cleft, which indeed demonstrate heightened mobility in NatB, offering supportive evidence. The robust correlation between the initiation of co-translational assembly interactions and a marked increase in cumulative interface energy strongly suggests that energy profiles hold promise in predicting co-translational assembly on a proteome-wide scale. This assertion is upheld by analyzing multiple yeast complexes, from various pathways, including the phosphofructokinase complex (PFK), anthranilate synthase complex (TRP), and Nucleoporin Nup85-Seh1 subcomplex. In these complexes, we were able to correlate energy profiles with the initiation of co-translational assembly interactions. Additionally, our analysis of the TRP complex and nucleoporin subcomplex highlighted the role of vectorial interface synthesis in directing the onset of co-translational interactions. Overall, our results emphasize the significance of energy profiling and 3D clustering of hotspots as powerful tools for predicting co-translational assembly interactions. These findings enhance our understanding of the dynamic molecular processes governing protein complex formation during translation, contributing to the broader knowledge of protein biophysics and cellular processes.

Markedly, numerous single nucleotide polymorphisms (SNPs) in human NatA and NatB complexes have been linked to neurodevelopmental and neurodegenerative disorders[15,33,64]. Our analysis delved into the conservation of identified interface hotspots and their relationship to these disease-associated SNPs. Noteworthy missense variants were found in highly conserved residues within both catalytic and auxiliary subunits, disrupting hotspot clusters. Notable examples include Naa10 D10G, L11R, and S37P (corresponding to D12, I13, and S39 in *S. cerevisiae*), all demonstrating high fluctuations compared to wildtype, disrupting the conformation of two N-terminal α-helices, predicted to cluster the strongest interface hotspots[24,61,62]. Our findings, coupled with in vivo mutation analyses, underscore the predictive capacity of MD-derived energy profiles concerning co-translational assembly interactions.

A meticulous exploration of the pathways that safeguard correct folding and assembly of the cellular proteome during synthesis offers the prospect of precisely predicting co-translational protein-protein interaction networks and interfaces. Such an endeavor could pinpoint essential amino acids crucial for upholding protein-protein associations within cells. The enduring conservation of conformational, energetic, and kinetic motifs positions them as promising foundations for devising novel interfaces. Prioritizing co-translational events sets the stage for advancing preventative healthcare strategies and identifying novel therapeutic targets for conditions marked by the accumulation of aberrant protein assemblies, including Parkinson's and Alzheimer's diseases[65,66].

## Methods

### PDB structures and AlphaFold2 modeling

For NatA and NatB, the catalytic–auxiliary complexes have experimentally solved structures in the PDB (6HD5 and 6HD7 for NatA, and 8BIP for NatB). Missing heavy atoms and loops were added with Modeller v.9.19[67], while hydrogen atoms were added with MolProbity[68]. Finally, the histidine residues' protonation state was revised and corrected manually when necessary.

We generated the complexes NatB, PFK, TRP, Nup85/Seh1 using AlphaFold-Multimer[31]. The NatB model was used for comparison and validation of the model against the PDB, running MD simulations for both, followed by the same extensive analysis (Supplementary Fig. 4; Supplementary Table 5).

## Structural alignment

All structure comparisons were made with the TM-align[38] algorithm, a method for sequence-independent protein structure comparisons. TM-align generates a structural alignment of the input structures and the TM-score, a structural similarity measurement (from 0 to 1, where 1 indicates fold identity). TM-score is more sensitive than RMSD to the global topology, and it is independent of protein size, problems common with RMSD comparison. For two protein structures of unknown equivalence, TM-align first generates an optimized residue-to-residue alignment based on structural similarity using heuristic dynamic programming iterations. An optimal superposition of the two structures built on the detected alignment, as well as the TM-score value, which scales the structural similarity, is returned.

Following strict statistics of structures in the PDB, TM-scores below 0.2 correspond to randomly chosen unrelated proteins, while those higher than 0.5 generally assume the same overall fold in SCOP/CATH. Additionally, TM-align provides a list of equivalent residues in both structures, which in this work, we considered as the common structural "core" shared by both structures.

## Molecular dynamics setup and protocol

Each PDB was pre-processed by the tleap program (AmberTools19 version) with the ff19SBonlysc[69] forcefield for proteins and GAFF[70] for ligands. OPC3BOX forcefield was used to soak the complex within a truncated solvated octahedron box with a minimum distance of 12 Å between any atom on the box edge, generating coordinates and parameter input files for molecular dynamics (MD) simulation[71]. Molecular Dynamics data was obtained by pmemd CUDA[72,73] version of Amber18 suite[74]. More details are provided in Supplementary Table 16.

The equilibration protocol consisted of an initial minimization, several steps of heating, and a gradual reduction of initial positional restraints. Equilibration of box volume was carried on at constant pressure while production was run at constant volume. Equilibration consisted of a total of 8 ns of MD with a time step of 1 fs (stages 1–8) and 1 ns of final 2 fs equilibration without constraints (stage 9). First, a minimization of water and added H atoms (stage 1); 1 ns of MD heating, using restraints on every atom at constant volume (NVT ensemble) (stage 2); Then 1 ns of MD at constant pressure (NPT ensemble) and constant temperature to adjust the density of water with full restraint on protein (stage 3); Then 1 ns of MD with lower restraints at NPT (stage 4); Then a second minimization of the side chains (stage 5); Then three stages of 1 ns MD at constant pressure with decreasing restraints on the backbone (stages 6–8); And finally a 2 ns unrestrained run at NTP (stage 9). Previous to the production run, hydrogen-mass repartitioning was applied to the input files to increase the time step to 4 fs. Then production data was obtained at NVT (stage 10). Periodic boundary conditions and Ewald sums (grid spacing of 1 Å) were used to treat long-range electrostatic interactions. During production runs, netCDF (Network Common Data Form) trajectory files were created.

Following molecular dynamics simulations, post-processing tools were used to obtain geometric, energetics, and other values.

## Free energy decomposition: MMPBSA

The free energy of binding for the bound complex was estimated using Mechanics Poisson-Boltzmann Surface Area (MMPBSA) embedded in the MMPBSA.py[49] module of AMBER18. MMPBSA was also used to compute the contributions from individual residues to the complex stability (free energy decomposition analysis). In total, 500 frames were post-processed from the trajectories, and the net complexation energy of the system was estimated through the following equation:

$$\Delta G_{binding} = \Delta G_{complex} - \Delta G_{partner1} - \Delta G_{partner2}$$

MMPBSA.py script computes the end-state free energy of the complexation process from an ensemble of representative structures

by summing up the solvation energies (polar and nonpolar) and the molecular mechanics (MM) energies. The contribution of polar solvation energy is calculated with the implicit solvent model (GB or PB), whereas the nonpolar part of the solvation energy is computed from the solvent-accessible surface area (SASA) difference between the complex and its free components.

Here, we used 500 snapshots collected of three equally simulation sections of 10 ns (spaced by 20 ps) of the production MD simulation to compute MMPBSA free energy of complex formation and the contribution of each residue (free energy decomposition energies) to the interface formation.

## Graphical visualization analysis tools

For trajectory visualization and molecular graphics and analysis figures, we used Visual Molecular Dynamics[75] (VMD) and UCSF ChimeraX[76]. ChimeraX was developed by the Resource for Biocomputing, Visualization, and Informatics at the University of California, San Francisco, with support from NIH R01-GM129325 and P41-GM103311.

## Molecular dynamics trajectory analysis

During the simulation, we used CPPTRAJ[77] to analyze the trajectory data and calculate root mean squared fluctuation (RMSF), root mean square deviation (RMSD) and monitor key distances, hydrogen bonds, etc.

RMSF computes the atomic positional fluctuations for any given set of atoms. The RMSF of a given atom i is calculated as $RMSF_i = \sqrt{\langle (x_i - \langle x_i \rangle)^2 \rangle}$. $x_i$ denotes the atomic positions of the i$^{th}$ atom and $\langle x_i \rangle$ denotes the average position for the selected snapshots.

RMSD performs the best fit of coordinates to a reference conformation and then calculates the deviation from each set of coordinates (i.e., snapshots) to the reference, with RMSD = 0.0 indicating a perfect overlap. RMSD is defined as:

$$RMSD = \sqrt{\sum_{i=0}^{N} \frac{\left[ m_i \cdot (X_i - Y_i)^2 \right]}{M}}$$

Where N is the number of atoms, $m_i$ is the mass of atom i, $X_i$ is the coordinate vector for target atom i, $Y_i$ is the coordinate vector for reference atom i, and M is the total mass. If the RMSD is not mass-weighted, all $m_i = 1$ and M = N.

## Docking and energy of interactions: pyDock package

To model the energetic details of molecular assemblies[78] and compute the complexation energy, we used pyDock[79] bindEy module[46], which applies an energy-based function to score protein-protein complexes using a unique combination of electrostatics, desolvation, and van der Waals terms.

Additionally, we use the OPRA module of pyDock, a propensity-based method to identify RNA-binding sites on proteins (Optimal protein-RNA area, OPRA[80]).

## Computation of protein electrostatics

To estimate the electrostatic contribution of each peptide, we used the APBS method (Adaptive Poisson-Boltzmann Solver) software package[81,82], a Poisson-Boltzmann equation solver for macromolecules. This computation maps and visualizes the electrostatic field values induced by a protein structure, which allows studying the potential role of electrostatics in a variety of activities of these structures.

## Graphs

All graphs were generated using PRISM GraphPad, gnuplot[83], and seaborn python library[84]. Unpaired two-sample t-test was performed using GraphPad Prism version 10.0.0 for Windows, GraphPad Software, Boston, Massachusetts USA, www.graphpad.com

## Strains construction

Homologous recombination: GFP-tagged strains and deletion strains were generated via homologous recombination and constructed according to Janke et al. (2004)[85].

For the GFP tag, a cassette containing the monomeric GFP gene and a Hygromycin resistance marker was amplified from the pYM44-mGFP plasmid. For gene deletions, a cassette containing only a selection marker was PCR amplified. All experiments were performed in the BY4741 strain background. *S. cerevisiae* strains used in this study are listed in Supplementary Table 12.

## CRISPR\Cas9

mCherry-tagged strains and point-mutations strains were generated via CRISPR\Cas9 as described in Ofri Levi et al.[55].

**gRNA design**. Each gRNA was designed as a 20 nt sequence flanked by an NGG sequence (Cas9 PAM). IDT's web server, https://eu.idtdna.com/site/order/designtool/index/CRISPR_SEQUENCE, was used with default settings to design gRNAs. gRNAs were selected according to IDT's gRNAs score, where higher scoring is less likely to exhibit off-target activity. We usually select gRNA with scores higher than 90. Each gRNA was designed such that successful recombination by the donor DNA will modify its recognition site and lower base pairing. This lowers the chances of further cleavages after a successful recombination event.

**Single-vector CRISPR cloning**. bRA66 (Addgene #100952) backbone was digested using BplI according to the manufacturer's protocol. The cut vector was purified through an agarose gel. gRNA sequences were designed with BplI flanking sequence at their 3′. Two complementary oligos were synthesized by IDT standard protocol. gRNA oligos (100 μM) were annealed in a thermocycler with the following program: 95 °C for 5 min, then a gradual decrease to 85 °C by 2 °C/sec, and then a gradual decrease to 25 °C by 0.1 °C per second. Annealed gRNA was ligated into bRA66 at the BplI site using standard T4 ligase protocol and transformed into DH5α competent bacteria. Positive clones were verified by sequencing.

**Transformation into *S. cerevisiae***. gRNA-containing bRA66 was transformed with a double-stranded 80-bp DNA fragment or PCR product as donor DNA into *S. cerevisiae* using LiAc. Plasmid and donor DNA were co-transformed to lower the background of colonies in which no double-strand break was made, compared to sequential transformation[55]. Transformants were grown on YPGal (1% yeast extract, 2% peptone, 2% galactose, and hygromycin at 200 μg/mL) plate to induce Cas9 expression. A sample without donor DNA was used to evaluate the gRNA efficiency and typically resulted in less than five colonies. Viable clones were isolated, and the genetic manipulation success was validated by western blot or sequencing of PCR products of the genomic region of interest.

## Yeast cultures

Yeast cultures were cultivated either in liquid yeast extract–peptone–dextrose (YPD)-rich medium or in synthetic dextrose (SD) minimal medium (1.7 g/L yeast nitrogen base with ammonium sulfate or 1.7 g/L yeast nitrogen base without ammonium sulfate with 1 g/L monosodium glutamic acid, 2% glucose and supplemented with a complete or appropriate mixture of amino acids) at 30 °C. For fatty acid supplementation, SD media was supplemented with 0.03% Myristic acid (Sigma, pre-dissolved in DMSO), 0.1% Tween-40 (Sigma), and 0.05% yeast extract.

## Yeast growth assay

Each strain was incubated overnight in 5 mL YPD in an orbital shaker (30 °C, 200 rpm). The starters were then diluted to $OD_{595} = 0.2$ and put in a 96-well plate in triplicates. The growth curves were obtained by measuring the $OD_{595}$ by TECAN infinite M Plex 200 pro every 9.5 min. Using the growth curve, the doubling time was calculated as seen in Haramati et al.[86].

## Immunoblotting

Samples were dissolved in a standard sample buffer and boiled at 95 °C for 5 min. Samples were separated on SDS–PAGE gels (4–12% gradient), transferred to polyvinylidene fluoride membranes, and immunoblotted. The following antibodies were used: polyclonal rabbit GFP antibody (antiserum from rabbit raised against YFP), Anti-HA [12CA5] recombinant mouse monoclonal antibody, and polyclonal antibody HA.11 was raised against the twelve amino acid sequence CYPYDVP-DYASL. Proteins were visualized by enhanced chemo-fluorescence reaction.

## Selective Ribosome Profiling (SeRP)

SeRP was performed mainly as described previously[30]. A method that enables the capture and characterization of co-translational interactions. SeRP enables global profiling of interactions of any factor with translating ribosomes in vivo. The method relies on flash freezing cells during their log phase, preserving active translation. Cell lysates are then treated with RNase to digest all RNA species in the cell except ribosome-protected mRNA fragments (ribosome footprints). The sample is then split into two parts; one part is directly used for isolation of all ribosome footprints (total translatome), and the second part is used for affinity-purification of the specific subset of ribosomes associated with a factor of interest (e.g. a quality-control factor), also termed the interactome. Then, the protected mRNAs are extracted and used for cDNA library generation, followed by deep sequencing. Comparative analysis of the total translatome and interactome samples allows for the identification of all ORFs that are associated with the factor, as well as the characterization of each ORF interaction profile at near-residue resolution. This profile reports on the precise engagement onset, dynamics, as well as ribosome speed.

**Purification of ribosome-nascent chains (RNCs) for SeRP**. RNC purification was performed according to previous work[6]. Briefly, ~800 mL of cell culture was grown to early log phase ($OD_{595}$ of 0.5), at 30 °C, in YPD. Cells were rapidly collected by vacuum filtration and then flash-frozen. Next, cells were lysed by cryogenic grinding in 900 μ of lysis buffer (20 mM Tris–HCl pH 8.0, 6 mM MgCl$_2$ 140 mM KCl, 0.1 mg/mL cycloheximide (CHX), 0.1% NP-401 mM PMSF, 2× protease inhibitors (cOMPLETE EDTA-free, Roche), 0.02 U/mL DNaseI (recombinant DNaseI, Roche), 20 mg/mL leupeptin, 10 mg/mL E-64, 40 mg/mL bestatin, 20 mg/mL aprotinin). Supernatants were divided into two: total (200 μL) and immunopurification (700 μL) translatome samples. Total samples were treated with 10 U of RNaseI per A260 nm of RNA for 25 min at 4 °C, then laid on 800 μL sucrose cushion (25% sucrose, 20 mM Tris-HCl pH 8.0, 140 mM KCl, 10 mM MgCl$_2$, 0.1 mg/mL CHX, 1× protease inhibitors) followed by centrifugation for 90 min at 75,000 rpm at 4 °C, then resuspended in lysis buffer. Immunopurification samples were digested by 10 U of RNaseI per A260 nm, together with ~50 μL of GFP-binder slurry for 25 min at 4 °C.

**cDNA library preparation for deep sequencing**. Library preparation was performed chiefly as described previously[6]. Briefly, RNA extraction was performed by warmed acid phenol (Ambion). For ribosomal footprints isolation, samples were run in 15% TBE–Urea polyacrylamide gel (Invitrogen) in 1× TBE (Ambion), then stained with SYBR gold (Invitrogen), then excised RNA fragments with a size between 5-25 nt. For end repair dephosphorylation, 2 μL 10× T4 polynucleotide kinase buffer without ATP (NEB), 1 mL murine RNase inhibitor, and 2 μL T4 polynucleotide kinase (NEB) were incubated at 37 °C for 1 h. For linker ligation, ~5 pmol RNA was mixed with 8 μL 50% sterile filtered PEG MW

8000, 2 μL DMSO, 2 μL 10× T4 RNA Ligase 2 buffer (NEB), 1 μL murine RNase inhibitor, 1 μL 1 mg/μL linker L1 and 1 μL truncated T4 RNA Ligase 2 (NEB) and incubated for 2.5 h at 37 °C or 23 °C. For reverse transcription, RNA was incubated in 10 μL 10 mM Tris-HCl pH 7.0, 1 μL 10 mM dNTP (NEB), 1 μL 25 linker L1′L20 and 1.5 μL DEPC H$_2$O at 65 °C for 5 min. Next, we added 4 μL 5× FSB buffer (Invitrogen), 1 μL murine RNase inhibitor, 1 μL 0.1 M DTT (Invitrogen), and 1 μL Superscript III (Invitrogen). Samples were incubated at 50 °C for 30 min. To hydrolyze RNA, 2.3 μL 1 N NaOH was added and samples were incubated at 95 °C for 15 min. For circularization, DNA was incubated in 15 μL 10 mM Tris-HCl pH 8.0 and 2 μL 10× CircLigase buffer (EPICENTRE), 1 μL 1 mM ATP, 1 μL 50 mM MnCl$_2$ and 1 μL CircLigase (EPICENTRE) at 60 °C for 2 h. Circularized DNA was used for PCR amplification followed by a quality control test and sequencing on a HiSeq 2000 (Illumina).

**Data analysis.** Sequenced reads were processed as previously described[6] using standard tools for trimming and genome alignment (Cutadapt, Bowtie2, Tophat2) as well as Python scripts adapted to *S. cerevisiae*. All analysis was done utilizing at least two independent biological replicates that were highly reproducible, as evaluated by Pearson correlation.

**Ratio-based enrichment profiles analysis.** We compared the RPM (reads per million mapped reads) interactome and translatome at each nucleotide along the ORFs. Pearson correlation analysis was done to test the reproducibility of the replicates. If a threshold of 0.6 was passed, genes were processed further. Only ORFs with > 64 reads in both subunit-bound and total translatome datasets were analyzed.

**RNA immunoprecipitation coupled with qPCR (RIP-qPCR)**
Our RIP-qPCR protocol was adapted from previously published methods[10,87,88], tailored to facilitate robust RNA immunoprecipitation coupled with quantitative polymerase chain reaction experiments. This technique is designed to capture and quantify RNA molecules specifically associated with target protein interactions.

**Yeast culture growth.** Cultures were cultivated in YPD overnight. 100 mL of fresh YPD was inoculated with cells from the overnight cultures, achieving an initial OD$_{595}$ of 0.2. These expression cultures were then grown at 30 °C and 200 rpm until they reached an OD$_{595}$ of 0.5–0.6. Upon achieving the desired growth phase, cells were harvested by centrifugation for 30 sec at 3000 × g, followed by resuspension in 0.5 mL YPD. After a 15 min recovery incubation, cells were flash-frozen in liquid nitrogen.

**Cell lysis and bead binding.** Upon freezing, cells were supplemented with 0.5 mL of frozen high-salt lysis buffer. This lysis buffer consisted of 20 mM Tris-HCl (pH 8.0), 140 mM KCl, 10 mM MgCl$_2$, 1 mM PMSF, 0.1% NP-40, cOMPLETE EDTA-free protease inhibitor (Roche), 0.02 U/uL DNaseI, and either 0.1 mg/mL CHX (Sigma-Aldrich) or 40 mM EDTA (Sigma-Aldrich). Cryogenic disruption was performed using the Cryo-Mill (Retsch) at 30 Hz for 2 min. The thawed lysate was transferred into 1.5 mL tubes and cleared at 20,000 × g and 4 °C for 3 min. The cleared supernatant was then incubated with equilibrated Protein A resin conjugated to antibodies against HA (mouse, IgG2a), along with 0.1 U/μL Ribolock (Invitrogen) to prevent RNA degradation. The lysate-bead mixture was incubated at 4 °C for 1 h with end-to-end mixing.

**Bead washing.** Following incubation, the beads underwent a series of washes through sequential centrifugation at 500 × g and 4 °C for 5 min. The beads were washed three times with 1 mL of wash buffer A (20 mM Tris-HCl, pH 8.0, 140 mM KCl, 20 mM MgCl$_2$, 0.1% NP-40, cOMPLETE EDTA-free protease inhibitor, and 0.1 mg/mL CHX) for 1 min each, employing end-to-end mixing. This was succeeded by two additional washes with wash buffer B (20 mM Tris-HCl, pH 8.0, 500 mM KCl,

20 mM MgCl$_2$, 0.01% NP-40, cOMPLETE EDTA-free protease inhibitor, and 0.1 mg/mL CHX) for 1 min and 4 min, respectively. The beads were subsequently resuspended in 500 μL of 10 mM Tris-HCl, pH 8.0.

**RNA extraction.** The extraction of RNA was initiated by the addition of 40 μL of 20% SDS and 750 μL of pre-warmed phenol-chloroform-isoamyl alcohol (PCI, 65 °C, Invitrogen). The mixture underwent incubation at 65 °C, 1400 rpm for 5 min, followed by rapid cooling on ice for 10 min. Subsequent centrifugation at 15,000 × g for 10 min separated the aqueous phase, which was then subjected to a second round of PCI extraction at room temperature for 5 min. Residual PCI was removed through diethyl ether washing, and the remaining solvent was evaporated in a Speedvac (Eppendorf).

**RNA precipitation.** For RNA precipitation, 3 M NaOAc (pH 5.5) was added to achieve a final concentration of 0.3 M. To the resulting precipitate, 2.5 μL of Glycoblue (Invitrogen) and an equal volume of isopropanol were introduced. This mixture was then subjected to overnight freezing at −80 °C. Subsequent centrifugation at 15,000 × g and 4 °C for 90 min generated a pellet, which was washed with 70% EtOH, dried in a Speedvac (Eppendorf), and eventually resuspended in 20 μL of 10 mM Tris-HCl (pH 8.0). Generally, RNA precipitations yielded 150–250 ng/μL of RNA.

**cDNA Synthesis and qPCR.** For cDNA synthesis, 500 ng of RNA was employed. The cDNA synthesis followed the instructions of the PrimeScript™ RT reagent kit (Takara), including an optional gDNA eraser step. Subsequently, real-time qPCR was carried out using the TaqMan™ Fast and Advanced Master Mix (Applied Biosystems) as per the manufacturer's recommendations. qPCR probes were procured from IDT Syntezza, with specifics indicated in Supplementary Table 13. The qPCR cycling conditions involved the QuantStudio 5 cycler (Applied Biosystems) with appropriate temperature profiles. Each experiment consisted of technical triplicates, and analysis was performed using QuantStudio analysis software (v1.5.1). Quality assessments were conducted, and replicates were adjusted as recommended by the software. Experiments with major technical discrepancies were excluded from analysis, and these instances are noted in the Source Data file.

Real-time qPCR was conducted using the PerfeCTa SYBR® Green FastMix (Quantabio) following the manufacturer's protocol. The qPCR reactions were carried out on the QuantStudio 1 cycler (Applied Biosystems) with the following cycling conditions: 95 °C for 30 sec, followed by 40 cycles of 95 °C for 3 sec and 60 °C for 30 sec. Images were captured during each cycle within the annealing/extension step. All qPCR assays were performed in technical triplicates to ensure reproducibility. Each experiment was subjected to comprehensive analysis using the QuantStudio analysis software (v1.5.1).

To maintain data integrity, rigorous quality assessment was performed using the QuantStudio software. If the software recommended, individual technical replicates were excluded from the analysis to mitigate any potential anomalies. Notably, experiments were considered valid only if two or more data points from technical replicates were omitted in cases where substantial deviations were detected. This stringent quality control criterion ensured the reliability of our qPCR.

**Imaging**
Cells were diluted after overnight growth in an SC medium to an OD$_{595}$ = 0.3 and were further incubated at 30 °C for 2 h. Fixated cells were imaged after one of four treatments: growth in optimal conditions at 30 °C for another hour, mild heat-shock at 37 °C for 1 h, 42 °C heat-shock for 10 min, supplementing the cells with cycloheximide to a final concentration of 0.1 mg/mL followed immediately by heat-shock at 42 °C for 10 min. Cells were fixed with 37% formaldehyde or were imaged live. High-sensitivity confocal imaging was performed on a

Nikon Eclipse Ti2 spinning disk. Images were acquired by using a Plan Apo λ 100x/ 1.45 Oil objective lens (Nikon) and a quad-band filter set and up to four diode laser lines (405 nm, 488 nm, 561 nm, 635 nm) with the NIS Elements Advanced Acquisition software (v.7.8.13.0, by Molecular Devices, was used for confocal imaging). Z stacks (0.2 μm steps) images were acquired for the 488 nm and 561 nm channels. All further processing of acquired images was performed with Imaris software. A maximal projection of ~7-9 Z stacks is shown. For foci quantification of subunits fused to GFP, both manual and automated ('Spots' Imaris tool feature) quantifications were performed. Approximately 150 cells were analyzed per sample with a total of three repetitions.

## N-terminomics

Hydrophobic Tagging-Assisted N-Termini Enrichment: HYTANE[58] was employed to investigate protein N-termini with enhanced selectivity and sensitivity. We utilized formaldehyde to label free N-terminal and lysine amino groups of proteins. Following labeling, the proteins were enzymatically digested into peptides using trypsin. Enrichment of N-terminal peptides was achieved through hydrophobic labelling of internal peptides using undecanal (50:1 w/w undecanal:protein), followed by reverse-phase chromatography (using OASIS-HLB), enabling the purification of N-terminal peptides at low acetonitrile concentrations. Subsequent mass spectrometry analysis allowed for the determination of the sequences of these enriched N-terminal peptides. The classification of peptides to NATs was done based on the start position and the sequence of the first amino acids of the peptide. NatA acetylates Ser-, Thr-, Ala-, Gly-, and Val- N termini following initiator Met (iMet) processing by methionine aminopeptidases or MetAPs[89–91], NatB acetylates Met residues followed by an acidic or Asn residue (MD/ME/MN/MQ[90,92–94]) or, in the case of NatC, Met followed by a hydrophobic residue (ML/MI/MF/MY/MK[95]). NatE substrates' sequence starts with MS/MT/MA/MV/MC.

Sample processing protocol: Wildype, *naa10Δ*, *naa15Δ*, mutant *NAA15* (R354E, R355E), and mutant *NAA15* (R354A, R355A) yeast cells were grown in appropriate SD media and plates to allow the selective growth of each strain. Single colonies of each strain were streaked onto 5 mL YPD media and grown overnight at 30 °C, 250 rpm. Cultures were then diluted 25-fold in 50 mL fresh YPD and grown until $OD_{595} = 1$. Cells were harvested by centrifugation at 4000 rpm for 5 min at room temperature. Yeast pellets were washed 3 times with ice-cold water and flash-frozen in liquid nitrogen and stored at −80 °C. Frozen yeast pellets were dissolved in 250 μL 1% sodium deoxycholate in 100 mM HEPES pH-8.0 and incubated at 95 °C for 10 min. Protein concentration was determined using the BCA method. The samples were then subjected to denaturation, reduction, and alkylation processes. The *naa10Δ*, *naa15Δ*, and *NAA15* mutant samples were labeled with light formaldehyde ($C_{12}H_2$), and wildtype samples were labeled with heavy formaldehyde ($C_{13}D_2$). Finally, equal protein amounts of light-labeled *naa10Δ/ naa15Δ*/mutant and heavy-labeled wildtype samples were mixed and subjected to N-terminal enrichment based on the HYTANE method using the protocol described in Hanna et al. (2023)[60]. All samples were prepared in a single biological replicate ($n = 1$).

Following enrichment and desalting, the samples were analyzed by tandem mass spectrometry using Thermo Q-Exactive Plus Orbitrap coupled to Easy nano-LC 1000 capillary HPLC. Enriched Terminal peptides were resolved on a homemade reverse phase capillary 30 cm long and 75 μm diameter, packed with 3.5 μm silica using ReproSil-Pur C18-AQ resin (Dr. Maisch GmbH). Elution was performed with a 120 min linear gradient of acetonitrile 5-28% (in 0.1% formic acid), followed by a 15 min wash with 95% acetonitrile (in 0.1% formic acid), with all flow rates set to 0.15 μL/min. MS was conducted in a data-dependent acquisition mode for positive ions in an m/z range of 300-1500, with resolution of 70,000 for MS1 and resolution of 17,500 for MS2. The ten most dominant ions selected from the MS1 scan were fragmented using high-energy collisional dissociation with 25 normalized collision energy.

Data processing protocol: Data analysis was performed using the Trans Proteomic Pipeline (version 6.1)[96]. Peptide search was done using Comet 2023.01 rev. 2[97]. The rest of the analysis was conducted for only N-terminal acetylated peptides as described in the data analysis section of the protocol in Hanna et al.[60].

Peptide scoring was done using PeptideProphet[98] at a false discovery rate of 1%. All data were searched against Uniprot Yeast proteome (UP000002311) containing 6090 sequences (downloaded April 2023) supplemented with known contaminant sequences and decoy sequences. Data are available via ProteomeXchange with the identifier PXD048082.

### Reporting summary

Further information on research design is available in the Nature Portfolio Reporting Summary linked to this article.

## Data availability

The translatomics data generated in this study have been deposited in the Sequence Read Archive repository under accession code: PRJNA1030163. Figure 1 also relies on raw data derived from the Gene Expression Omnibus repository: GSE116570. Figure 2 also relies on raw data derived from GSE93830. Figure 4 also relies on raw data derived from GSE116570, PRJEB46361, and PRJEB50305. The MS data generated in this study have been deposited in the ProteomeXchange Consortium with identifier PXD048082. All other data are available from the corresponding authors upon request. Source data are provided with this paper.

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

## Acknowledgements

We would like to thank all the lab members for fruitful discussions, and especially Rawad Hanna for his contribution to N-terminomics analysis. We thank the Technion's Lorry I. Lokey Interdisciplinary Center for Life Sciences and Engineering for the core facility. We are grateful for the professional services provided by Nitsan Dahan and Yael Lupu-Haber from the Microscopy Imaging and BioAnalysis unit of the LS&E Infrastructure Center of the Technion, Israel. We also thank the Technion

Smuler Proteomic Center and especially Ilana Navon for their ongoing support and help with MS measurements. This work was funded by the European Research Council Starting Grant 2031817 (A.S.) and the Israeli Science Foundation grants 2106/20 (A.S.) and 1515/23 (O.K.). This work was also partially supported by grant number PID2022-143215OB-I00, funded by MCIN/AEI/10.13039/501100011033/ FEDER, UE (J.F.R).

## Author contributions

Conceptualization, J.V., J.F.R., F.G. and A.S.; Methodology, J.V., H.B.Y., H.B.A.Z., N.C., O.K., F.G. and A.S.; Software, F.G.; Validation, J.V., H.B.Y., H.B.A.Z., N.C., F.G. and A.S.; Formal Analysis, J.V., H.B.Y., H.B.A.Z., N.C., F.G. and A.S.; Investigation, J.V., H.B.Y., H.B.A.Z., F.G. and A.S.; Writing – Original Draft, J.V., F.G. and A.S.; Writing – Review & Editing, J.V., H.B.Y., H.B.A.Z., F.G. and A.S.; Funding Acquisition A.S.; Resources, H.B.Y. and A.S.; Supervision, O.K. and A.S.

## Competing interests

The authors declare no competing interests.
