## [Peer Review File · Nature Communications]

Diverging co-translational protein complex assembly pathways are governed by interface energy distributionREVIEWER COMMENTS

Reviewer #1 (Remarks to the Author):

In this manuscript, Venezian et al. combined SeRP, imaging, and N-terminomics with MDs to uncover co-translational assembly pathways of selected yeast proteins as principal examples. They identified so-called “hotspots”, a few clustering residues characterized by high binding energy, which serve as initiating co-translational assembly interactions upon exposure at the ribosome exit tunnel to build the entire interface assembly. Importantly, in vivo hotspot mutations disrupted co-translational assembly and resulted in aggregation of orphaned subunits whose proper folding depends on co-trans. assembly. Based on their results, the authors proposed a general model utilizing the knowledge of the interface energy distribution that could serve as a strong predictor of co-translational assembly schemes. In addition, they identified mutations in human homologues of their proteins of interest disrupting these putative hotspot clusters, which could underlie neurodevelopmental and neurodegenerative diseases associated with mutations in these proteins.

Considering the broad portfolio of techniques used in this study, I must admit that I only have a deep understanding of SeRP, RIP, CRISPr and cell biology assays but not of the energy landscape and MD simulations. Nonetheless, the parts that heavily depend on these techniques and that I cannot expertly judge sound very logical to me. Overall, I found this study very intriguing, well thought through, all experiments very well executed and controlled, in one word – complete. Complete and very useful for the gene expression field. Below I only list a few minor comments.

Fig. 1g-h. Statistical significance for lower temperatures looks a bit surreal to me.

Fig. 1d. Color-coding of Naa10 and 20 – the shades of green – are very hard to distinguish.

Fig. 2. For better clarity, instead of ScNatB etc. (or in addition to it), please use the protein names. In general, the description of all figures uses super small letter sizes, so a magnifying glass should be called upon to return back to its service to save eyes.

The last two paragraphs of the “Assembly hotspots...” chapter lack references to relevant figures.

RIP-qPCR in Fig. 3 – my question, just a question is whether or not SeRP could be used instead of RIP to detect similar differences?

Fig. 4. Please add the explanation of the vertical dashed line (onset of co-assembly determined in Nat 2018) underneath each plot after “Cumulative”.

Page 5, first column. Fig. 3i should be Fig. 3i-j, I think.

Page 5, second column. Fig. 3c should be Fig. 4c, I think.

Fig. 4. “Core-domain (aa 300-900)” should be depicted in the panel C.

Page 6, 1st column. Po-translational should be post-translational, I hope :)

Thank you for giving me the opportunity to review this article. Leos S. Valasek

Reviewer #2 (Remarks to the Author):

Venezian et al. investigate the characteristics of co-translational protein complex assembly. To do so, they use known examples and perform MD simulations to obtain insights into the hierarchy of these interactions as well as the amino acid residues that may drive these interactions. After identifying amino acids that may drive co-translational interactions, they authors validate their results by mutating them in the endogenous genes, followed by phenotypic assays, demonstrating that the amino acids obtained from MD simulations are indeed the drivers of co-translational protein complex assembly.

This is a nicely executed study on an important problem. Although many proteins use this kind of assembly, it is rarely studied, partially as there are only few approaches. These limitations, make this paper even more important.

I have only a few suggestions, but they need to be addressed. The biggest problem is that many figure references are messed up and wrong and that many figures are not referenced in the text. I suspect that the paper was rearranged, but the figure numbers were not updated. This makes the evaluation of the paper labor intensive and difficult. As many figures are not referenced in the text, their purpose and how they help the story is unclear.

Specific points.

1. Fig. 2a vs 2b and Fig. 2c vs 2d. The pattern of the amino acids with the largest free energy change (labeled in red) corresponds to the description of the authors (the red dots are more evenly distributed in recruited subunits and more clustered in co-translational subunits). Although this pattern is somewhat visible by eye, the presented data would be more convincing if a statistical analysis would be presented.
2. Fig. 3g. Also here, no statistical test or replicates were performed. At the current stages, this makes the analysis descriptive and anecdotal.
3. What is the statistical test, performed in Fig. 1g? This is not obvious, as the figure legends of Fig. 1 do not correspond to the panels shown in Fig. 1.
4. They authors mention that mutations in human NATs that cause neurodevelopmental diseases that can disrupt hotspots for interaction. Why did the authors not mutate these residues in yeast to assess potential phenotypes or at least model the effects using simulations? At least the modeling should be performed. It would be useful to show the alignment of human and yeast sequence.
5. Figures 2g-k are not referred to in the text. Fig. 2i is referred to in the text, but the text does not fit to the figure shown.
6. Fig. S1, S2, S4a-d are not referred to in the text. Fig. S4 says that it is related to Fig. 4, but Fig. 4 does not deal with NatA or NatB.
7. The text for Fig. 4c mentions a Tm score, but the figure is unrelated. The minimal region of Pfk2 was never introduced.
8. Currently Fig. 4 shows that their strategy of finding interaction hotspots can be expanded to other candidates. This is an important section and should be expanded in the text and in the figures to make things clearer for the reader. For example, there is only a small paragraph on TRP, but it is unclear what is learned from it. Please expand or delete.

Also, the paragraph on the nucleoporin needs to be expanded or deleted. It is currently written 'in code' for insiders.
9. The model figure (4d) was not mentioned in the text and seems unrelated to the rest of the paper. It is unclear what the purpose of Fig. 4b and 4c is.

Reviewer #3 (Remarks to the Author):

In their manuscript (NCOMMS- 23-48035) Shiber and co-workers use extensive modeling and analyses, and experimental measurements to investigate diverging co-translational assembly pathways of NATs. The study is based on extensive and well-designed set of all-atom simulations of systems. The study is detailed, and findings are well supported.

The all-atom molecular dynamics trajectories and analyses enabled authors to capture a small set of “hotspot” residues that play a key role in co-translational interactions and complex assembly. What makes author’s approach powerful is a close connection between modeling and supported.

These findings have implications for future studies of co-translational events and protein-protein-protein interactions often implicated in disease and will be of great interest to the readers of Nature Communications.

The manuscript can be published mostly as is, though it would be helpful to edit the text (and figure captions) for clarity and simplification, including reducing the size of some paragraphs as well as perhaps moving some figure panels to SI.

Reviewer #4 (Remarks to the Author):

The manuscript dissects the cotranslational assembly pathways of the NatA and NatB acetyltransferase complexes. Initial selective ribosome profiling (seRP) shows that the assembly of these homologous complexes uses two distinct routes. In the case of NatA, the catalytic subunit Naa10 is fully folded and assists folding of the auxiliary subunits Naa15, whereas for NatB the auxiliary subunit Naa25 is translated and assists folding of the catalytic subunit Naa20.

The authors apply extensive modeling studies based on known CryoEM structures and AlphaFold models to derive hypotheses explaining this behavior. This includes MD simulations and MM-PBSA energy profiling that identify hotspot regions and residues that mediate these co-translational interactions and onset of these interactions during translation. The predictions are experimentally tested in yeast carrying point mutations in both NatA subunits, with growth assays, assays of aggregate formation and N-terminome profiling as functional readouts.

In the last section, similar calculations are performed for other complexes with previously identified cotranslational interactions between partner subunits, demonstrating that this computational approach can be applied more generally to predict sites and onset of cotranslational interactions.

Overall, the manuscript provides a compelling story combining computational modeling and experimental validation. The computational methods appear solid, but I am not particularly familiar with these. The experimental section requires some clarifications and the manuscript as a whole a detailed proof-reading for clarity and consistency. In particular, several figure references appear to be inaccurate.

Detailed comments:

Page 2, left column, about middle: “We simulated the Cryo-EM structures of NatA and NatB.” – I assume you modeled the structures of NatA and NatB based on the cryo-EM structures?

Page 4: “For NatA subunits, binding of TRiC/CCT was also hardly detected, interacting only with a very short segment in the catalytic subunit (residues 103- 109)”. – Please show data as supplementary figure.

The NAA15 R354E/R355E double mutant has a similar effect as naa10delta, implying that NatA is effectively depleted. What happens to NAA10 in this mutant –is it soluble, or co-precipitating with in the aggregating NAA15 or is it degraded?

Functional readout by N-terminome analysis:

Figure 3 j: what is plotted on the x-axis: log 10 of the summed intensity of both channels?

Some NatA substrates are acetylated in the naa15delta. How is this rationalized? Are there specific subsets that require NAA15? What are the N-terminal sequence motifs of these populations?

In both mutants: Is this decrease in acetylation mirrored by an increase in the acetylated counterpart, or are these associated proteins degraded due to missing acetylation?

Add a table summarizing your identifications with peptide sequence, modification, avr. Fold change, significance score, and individual quantifications in each replicate.

Methods:

Enrichment of N-termini is not precise. Was Undecanal tagging really employed in the first step for modification, i.e. was undecanal used for positive selection of N-terminal peptides?

Similarly, the methodology used and cutoffs applied for data analysis must be better defined. Extend this section to match information provided in repository upload.

Raw data is accessible, but only via Nature Communications Reporting summary file. Add repository dataset identifier and reviewer access information to the methods section as standard practice for proteomeXchange-deposited data.

References to figures:

Page 3: left column, about middle: A section mentions and discusses findings in the crystal structure with reference to Fig. 3g – this does not appear to match the information provided in the actual Fig. 3g.

Figure 3 c/d, references to this figure in page 4 top left paragraph: Fig 3.c and d appear changed (or panels swapped)

Page 5, discussion of N-termini: check reference to figure panels (3i referring to ratios instead of scheme, no reference to 3j)

References 6 and 28 are identical

Point-by-point response

Nature Communications manuscript NCOMMS-23-48035

We thank the reviewers for their thoughtful and constructive comments, and the editor for inviting us to submit a revised manuscript. We have prepared a revised manuscript that carefully addresses all concerns.

In the following, we are providing the full text of the reviews and our response (bold and italicized) to each concern.

REVIEWER COMMENTS

Reviewer #1 (Remarks to the Author):

In this manuscript, Venezian et al. combined SeRP, imaging, and N-terminomics with MDs to uncover co-translational assembly pathways of selected yeast proteins as principal examples. They identified so-called “hotspots”, a few clustering residues characterized by high binding energy, which serve as initiating co-translational assembly interactions upon exposure at the ribosome exit tunnel to build the entire interface assembly. Importantly, in vivo hotspot mutations disrupted co-translational assembly and resulted in the aggregation of orphaned subunits whose proper folding depends on co-trans. assembly. Based on their results, the authors proposed a general model utilizing the knowledge of the interface energy distribution that could serve as a strong predictor of co-translational assembly schemes. In addition, they identified mutations in human homologs of their proteins of interest disrupting these putative hotspot clusters, which could underlie neurodevelopmental and neurodegenerative diseases associated with mutations in these proteins.

Considering the broad portfolio of techniques used in this study, I must admit that I only have a deep understanding of SeRP, RIP, CRISPR, and cell biology assays but not of the energy landscape and MD simulations. Nonetheless, the parts that heavily depend on these techniques and that I cannot expertly judge sound very logical to me. Overall, I found this study very intriguing, well thought through, and all experiments very well executed and controlled, in one word – complete. Complete and very useful for the gene expression field. Below I only list a few minor comments.

Fig. 1g-h. Statistical significance for lower temperatures looks a bit surreal to me.

We thank the reviewer for their comment and therefore added in the figure legend an indication for the statistical test performed and the number of points on which it was applied.

Fig. 1d. Color-coding of Naa10 and 20 – the shades of green – are very hard to distinguish.

As per the reviewer's request, we modified the colors according to the rest of the figures, so it would fit the color scheme and will be easier to differentiate.

Fig. 2. For better clarity, instead of ScNatB etc. (or in addition to it), please use the protein names. In general, the description of all figures uses super small letter sizes, so a magnifying glass should be called upon to return back to its service to save eyes.

We thank the reviewer and updated all graphs and figures to indicate the protein names, removing the prefix “Sc” (ScNatX -> NatX). Also, we increased all font sizes for better legibility.

The last two paragraphs of the “Assembly hotspots...” chapter lack references to relevant figures.

We thank the reviewer for bringing this to our notice and added the proper references.

RIP-qPCR in Fig. 3 – my question, just a question is whether SeRP could be used instead of RIP to detect similar differences?

RIP-qPCR provides a more quantitative assessment of interactions versus SeRP, though missing the per-codon resolution of interaction. This enabled us to accurately compare the impact on mRNA enrichment levels of the various mutants under EDTA versus CHX treatments.

Fig. 4. Please add the explanation of the vertical dashed line (onset of co-assembly determined in Nat 2018) underneath each plot after “Cumulative”.

We thank the reviewer and added an explanation of the dashed line in the cumulative graphs to the figure legend.

Page 5, first column. Fig. 3i should be Fig. 3i-j, I think.

Figure references were edited accordingly, to better fit the text.

Page 5, second column. Fig. 3c should be Fig. 4c, I think.

We thank the reviewer and corrected the references accordingly.

Fig. 4. “Core-domain (aa 300-900)” should be depicted in the panel C.

We thank the reviewer and have updated Fig. 4c and its legend. It is now split into 4b and 4c to better indicate the N-terminal (glyoxalase-like) domain versus the other (phosphofructokinase) domain of the PFK subunits. We have also updated the relevant segment in the results, indicating in detail the PFK subunits’ domain organization: “Pfk2 has a distinct 50-amino acid IDR (aa 146-198). Notably, this IDR serves as a long linker between two domains that display high similarity in both subunits: the N-terminal glyoxalase-like domain and the middle phosphofructokinase domain. This extended IDR allows the middle phosphofructokinase domain to adopt a 180° flipped position relative to its N-terminal domain (Fig. 4c). This enables an asymmetric interface formation leading to an asymmetric co-translational assembly pathway. Thus, similar to the NATs case, energy profiling allows for onset prediction, where structural features are too homologous for distinction.”.

Page 6, 1st column. Pot-translational should be post-translational, I hope :)

We thank the reviewer and corrected it to “post-translational”.

Thank you for giving me the opportunity to review this article. Leos S. Valasek

Reviewer #2 (Remarks to the Author):

Venezian et al. investigate the characteristics of co-translational protein complex assembly. To do so, they use known examples and perform MD simulations to obtain insights into the hierarchy of these interactions as well as the amino acid residues that may drive these interactions. After identifying amino acids that may drive co-translational interactions, they authors validate their results by mutating them in the endogenous genes, followed by phenotypic assays, demonstrating that the amino acids obtained from MD simulations are indeed the drivers of co-translational protein complex assembly.

This is a nicely executed study on an important problem. Although many proteins use this kind of assembly, it is rarely studied, partially as there are only few approaches. These limitations, make this paper even more important.

I have only a few suggestions, but they need to be addressed. The biggest problem is that many figure references are messed up and wrong and that many figures are not referenced in the text. I suspect that the paper was rearranged, but the figure numbers were not updated. This makes the evaluation of the paper labour intensive and difficult. As many figures are not referenced in the text, their purpose and how they help the story is unclear.

Specific points.

1. Fig. 2a vs 2b and Fig. 2c vs 2d. The pattern of the amino acids with the largest free energy change (labelled in red) corresponds to the description of the authors (the red dots are more evenly distributed in recruited subunits and more clustered in co-translational subunits). Although this pattern is somewhat visible by eye, the presented data would be more convincing if a statistical analysis would be presented.

We thank the reviewer for their insightful suggestion, and we applied Kolmogorov-Smirnov test to the cumulative energy functions to determine differences and significance in a more precise manner, providing an exact calculation. See revised manuscript, legend of Fig. 2: "The cumulative distributions were found to be significantly different based on two-sample Kolmogorov-Smirnov test with a p-value = 1.688e-09 for the catalytic subunits and p-value = 2.313e-07 for the auxiliary subunits".

2. Fig. 3g. Also here, no statistical test or replicates were performed. At the current stages, this makes the analysis descriptive and anecdotal.

We thank the reviewer for pointing out the ambiguity in the figure and its legend. Growth assays were performed in triplicates and statistical significance was calculated. We have revised the figure and legend to indicate this more clearly:

Fig. 3g: "Mutation analysis of interface hotspots reveals their significant impact on growth. The minimal doubling time of wildtype and the mutated strains, extrapolated from growth curves (see Fig. S3b corresponding to the small graph at the top left; mean \pm SD, n = 3, unpaired two-sample t-test)."

Fig. S3b: "Growth curves analysis of the indicated mutated strains. Averaged OD₅₉₅ of three biological replicates, each representing a growth curve of one strain. The degree of experimental variation (standard error) is shaded in the corresponding color."

3. What is the statistical test, performed in Fig. 1g? This is not obvious, as the figure legends of Fig. 1 do not correspond to the panels shown in Fig. 1.

We thank the reviewer and now the statistical test performed (unpaired two-sample t-test) is properly indicated in the legend.

4. The authors mention that mutations in human NATs that cause neurodevelopmental diseases that can disrupt hotspots for interaction. Why did the authors not mutate these residues in yeast to assess potential phenotypes or at least model the effects using simulations? At least the modelling should be performed. It would be useful to show the alignment of human and yeast sequence.

We thank the reviewer and have now performed molecular dynamics simulations to assess potential phenotypes in the human Naa10. The MD results predict the mutants will significantly impact the conformation of α -helices harboring the highest cluster of human Naa10 interface hotspot residues. We have revised the manuscript, and supplementary figure S5 accordingly:

Figure S5: Human Naa10 disease mutants' MD structural analysis reveals a significant impact on alpha helices harboring predicted interface hotspots.

(a) A model of human Naa10 heterodimer generated by AlphaFold. The catalytic subunit Naa10 is in green, and the auxiliary subunit is in tan.

(b) Human Naa10 from the solved structure PDB: 6C9M 2, highlighting complex interface hotspots in red. Naa10 complex interface energy contribution per residue in a range of -2 (red) – 0 (grey) $\Delta\Delta G$ [kcal/mol], computed from 300 ns MD simulations, using 1000 frames from the last 20ns, with pyDock bindEY. α -helices at positions: aa 8-20 and aa 28-37, centered, identified as clustering residue hotspots contributing the most energy to the interface.

(c) Wildtype human Naa10 highlighting disease mutants D10G, L11R and S37P (stick representation). All residues are located at the two highly energetic helices.

(d-e) Free human Naa10 subunit thermostability in wildtype (d) compared to disease mutants (e) D10G, L11R, and S37P. Conformational changes predicted by MD simulations at 30°C, over 300 ns timeframe. Timepoint 0 ns of the simulations as displayed in (c). Only the frame at 300 ns is shown for all. The mutated proteins were obtained by replacing the wildtype residues and equilibration before running production. The MD simulations show the mutants impact the conformation of the two α -helices harboring many hotspots.

(f-g) RMSD for wildtype, D10G, L11R, and S37P computed for the two α -helices. The RMSD boxplots (g) indicate that the conformation of the wildtype is maintained during the simulation while the mutants change conformation relative to the starting point (unpaired two-sample t-test).

(h) RMSF of the wildtype and the mutants, per residue, along the ORF.

(i) RMSF boxplots of the entire protein or the indicated segments, demonstrating the higher fluctuations of the mutants at the two α -helices (unpaired two-sample t-test).

Revised manuscript text:

“Conservation analysis revealed that missense mutations in human NATs, causing neurodevelopmental and neurodegenerative diseases, can disrupt putative hotspots clusters (Fig. S5). For example, in the catalytic subunit of human NatA, D10G and L11R (D12 and I13 in *S. cerevisiae*) have been linked to several neurodevelopmental pathologies⁶⁰. These missense variants are far removed from the catalytic site; however, they caused a significant reduction in the observed catalytic activity. Simulating the mutated proteins compared to the wildtype human Naa10 revealed their impact on its fold, causing a shift in the conformation of two α -helices at the N-terminal harboring most of the interface energy (Fig. S5). Thus, mutations can impair the ability to form the interface with the auxiliary subunit, thus inhibiting the complex’s substrate binding and catalytic activity. Similarly, the conserved Naa10 S37 (S39 in *S. cerevisiae*) mutation to proline was found to cause Ogden syndrome, impairing complex assembly and catalysis²⁴. The alpha-helix harboring S37 encompasses six hotspots, to a total energy of -29.4 kcal/mol (Fig. S5b, Table S6d), showing S37P destabilizes the alpha helix, disrupting assembly and catalysis. Several other missense variants in conserved residues, such as Naa15 K338D^{61,62} (K358 in *S. cerevisiae*), all involved in developmental delay and microcephaly, can cause similar defects in assembly, as they disrupt predicted hotspots clusters.”

5. Figures 2g-k are not referred to in the text. Fig. 2i is referred to in the text, but the text does not fit to the figure shown.

We thank the reviewer and corrected the manuscript so now all figures are referred to in the proper context and with a fitting descriptive legend.

6. Fig. S1, S2, S4a-d are not referred to in the text. Fig. S4 says that it is related to Fig. 4, but Fig. 4 does not deal with NatA or NatB.

We thank the reviewer for pointing out this ambiguity. We have revised the text accordingly. In particular, Fig. S4 demonstrates the robustness of AlphaFold-Multimer modeling for downstream MD analysis, including interface hotspots recognition. This is shown by comparing NatB cryo-EM derived analysis to one based on AlphaFold-Multimer modeling. It was then implemented for phosphofructokinase complex (PFK), anthranilate synthase complex (TRP), and nucleoporin Nup85-Seq1 subcomplex in yeast. We have revised the manuscript accordingly: “The robustness of this methodology was demonstrated for NatB (Fig. S4), comparing its cryo-EM derived analysis to one based on AlphaFold-Multimer modeling.”

7. The text for Fig. 4c mentions a Tm score, but the figure is unrelated. The minimal region of Pfk2 was never introduced.

We thank the reviewer and have now revised Fig. 4c and related text to introduce Pfk2 minimal region and its different orientation. See revised Fig. 4b,c and related text in the answer to comment 8 below.

8. Currently Fig. 4 shows that their strategy of finding interaction hotspots can be expanded to other candidates. This is an important section and should be expanded in the text and in the figures to make things clearer for the reader. For example, there is only a small paragraph on TRP, but it is unclear what is learned from it. Please expand or delete. Also, the paragraph on the nucleoporin needs to be expanded or deleted. It is currently written 'in code' for insiders. *We thank the reviewer and have now expanded the segment on TRP, as well as added a complimentary figure. We have similarly expanded the text and figure for PFK and the text regarding the nucleoporin subcomplex. See the revised text and figure:*

Text: "In this study, we aimed to assess the predictive capability of energy profiles concerning the onset of co-translational interactions in specific protein complexes. Our investigation expanded to include the phosphofructokinase complex (PFK), anthranilate synthase complex (TRP), and nucleoporin Nup85-Seh1 subcomplex in yeast. Prior studies using SeRP revealed the onset of co-translational interactions for these complexes^{6,8}, so we could correlate directionality and onset with energy profiles (Fig. 4a). Due to crucial segments missing in the solved structures, we employed AlphaFold-Multimer³¹ to model the complete complexes, on which we ran MD simulations followed by MMPBSA analysis. The robustness of this methodology was demonstrated for NatB (Fig. S4).

The PFK complex is composed of alpha (Pfk1) and beta (Pfk2) subunits with 50% sequence identity and a TM-score of 0.79 (Fig. 4b,c). Despite sharing high structural similarity, these subunits exhibit distinct onset of co-translational interactions. For Pfk1 onset occurs when the first ~200 residues of Pfk1 nascent-chain are exposed. In Pfk2, onset occurs much later, when ~450 amino acids are synthesized, and enrichment levels fluctuate much more compared to Pfk1, until the end of synthesis⁶. Notably, the energy profiles closely correlate with these onset points. Pfk1 displayed a prominent clustering of hotspots in its first 200 residues, whereas Pfk2 demonstrated a more even distribution of hotspots along its first 450 residues, and overall along the ORF (Fig. 4a). The differences in the energy profiles lead us to investigate the subtle structural differences in the minimal regions, synthesized prior to co-translational assembly onset (Fig. 4b,c). Pfk2 has a distinct 50-amino acid IDR (aa 146-198) that serves as a long linker between two domains which display high similarity in both subunits: the N-terminal glyoxalase-like domain and the middle phosphofructokinase domain. This extended IDR allows the middle phosphofructokinase domain to adopt an 180° flipped position relative to its N-terminal domain (Fig. 4c). This enables an asymmetric interface formation leading to an asymmetric co-translational assembly pathway. Thus, like the NATs case, energy profiling allows for onset prediction, where structural features are too homologous for distinction.

The TRP heterodimer consists of the highly conserved Trp2 and Trp3 subunits. Energy profiling exhibited a robust correlation with the onset of interactions in Trp3, with significant hotspot clustering observed just prior to the onset (Fig. 4a). However, for Trp2, despite detecting hotspot clustering around 350 residues, no co-translational interactions were observed. This discrepancy can arise from the intricate fold of Trp2. Its interface can only form after the synthesis of its C terminus as adjacent β -strands are separated by a ~150 aa gap (β -strand in position 299-303 aa is connected to β -strand in position 445-449 aa, for example; Fig. 4d). Thus, co-translational interface formation cannot occur, as segments that are synthesized far apart are co-dependent on each other for folding and stability.

Regarding the nucleoporin subcomplex, energy profiling of Seh1 demonstrated an evenly-distributed interface energy along the ORF, encompassing its extreme C terminus. This distribution coincided with the lack of interactions detected during the synthesis of this subunit. In contrast, Nup85 displayed a strong clustering of hotspots in its extreme N terminus, contributing over 80% of its interface interaction energy. Nevertheless, co-translational complexation was observed only after the synthesis of the second hotspots cluster. The first cluster region is highly flexible. We hypothesize that the first cluster (aa 47-95) can only stably fold upon the synthesis of the second cluster (approximately aa 450-550), as they are closely interacting, including several hydrogen bonds, despite their distance in

the linear sequence (Fig. 4e). Thus, like Trp2, only after the synthesis of the second cluster can the entire interface form. In contrast to Trp2, this second cluster is exposed at the ribosome exit tunnel at aa ~565, when approximately 200 aa of Nup85 are yet to be synthesized, allowing for co-translational assembly interactions.”

Fig. 4b-e

b, AlphaFold-Multimer predicted structures of the PFK complex, composed of alpha (Pfk1) and beta (Pfk2) subunits. Comparison of Pfk1 and Pfk2 demonstrates their high structural similarity, except for a unique 50 aa IDR (dark blue) in Pfk2 as determined by TM-Align. Left: Aligned aa 1-150 with RMSD of 0.8 Å between 62 pruned atom pairs (out of 105). Right: Aligned the rest of the protein with RMSD of 0.5 Å between 707 pruned atom pairs (out of 764). Highlighted in a darker shade are the aligned regions.

c, Minimal regions, prior to co-translational complex assembly (in darker color) versus the post-minimal regions (in lighter color). Pfk2’s unique IDR allows its N-terminal domain to adopt a distinct position compared to Pfk1 in the complex (right, upper panel vs. lower panel). This structural divergence may underlie the variation in Interface energy contribution: Pfk1 displays a prominent clustering of hotspots in its first 200 residues (dark green), correlating to the onset of co-translational assembly interactions, whereas in Pfk2 most of the hotspots are in the first 450 amino acids (dark blue), correlating to its onset of co-translational assembly interactions.

d, AlphaFold predicted structure of Trp2, rainbow coloured from N- to C-terminal, displaying its intricate fold where adjacent β-strands are separated by a ~150 aa gap. Highlighted are β-strands aa 299-303 and aa 445-449, allowing interface domains to fold only after C-terminal segments’ synthesis.

e, AlphaFold-Multimer predicted structures of NPC subcomplex Nup85-Seh1 (left) and Nup85 alone (right). Nup85’s two hotspot clusters are highlighted in red (aa 47-95) and blue (aa 450-550), surface display. Zoom-in on intramolecular hydrogen bonds between the first and second hotspot clusters areas suggest their stabilization is co-dependent, only allowing for co-translational complex assembly interactions after the second cluster’s synthesis.

9. The model figure (4d) was not mentioned in the text and seems unrelated to the rest of the paper. It is unclear what the purpose of Fig. 4b and 4c is.

We thank the reviewer and have expanded and clarified Fig. 4b and 4c and related text. See above answer to comment 8. The model figure is also now referenced in the text and is explained better.

Reviewer #3 (Remarks to the Author):

In their manuscript (NCOMMS- 23-48035) Shiber and co-workers use extensive modelling and analyses, and experimental measurements to investigate diverging co-translational assembly pathways of NATs. The study is based on extensive and well-designed set of all-atom simulations of systems. The study is detailed, and findings are well supported.

The all-atom molecular dynamics trajectories and analyses enabled authors to capture a small set of “hotspot” residues that play a key role in co-translational interactions and complex assembly. What makes author’s approach powerful is a close connection between modelling and supported.

These findings have implications for future studies of co-translational events and protein-protein interactions often implicated in disease and will be of great interest to the readers of Nature Communications.

The manuscript can be published mostly as is, though it would be helpful to edit the text (and figure captions) for clarity and simplification, including reducing the size of some paragraphs as well as perhaps moving some figure panels to SI.

We thank the reviewer, and we have accordingly edited the text and figure legends for clarity and simplification, as can be seen in the revised manuscript.

Reviewer #4 (Remarks to the Author):

The manuscript dissects the co-translational assembly pathways of the NatA and NatB acetyltransferase complexes. Initial selective ribosome profiling (SeRP) shows that the assembly of these homologous complexes uses two distinct routes. In the case of NatA, the catalytic subunit Naa10 is fully folded and assists folding of the auxiliary subunits Naa15, whereas for NatB the auxiliary subunit Naa25 is translated and assists folding of the catalytic subunit Naa20.

The authors apply extensive modelling studies based on known CryoEM structures and AlphaFold models to derive hypotheses explaining this behaviour. This includes MD simulations and MM-PBSA energy profiling that identify hotspot regions and residues that mediate these co-translational interactions and onset of these interactions during translation. The predictions are experimentally tested in yeast carrying point mutations in both NatA subunits, with growth assays, assays of aggregate formation and N-terminome profiling as functional readouts.

In the last section, similar calculations are performed for other complexes with previously identified co-translational interactions between partner subunits, demonstrating that this computational approach can be applied more generally to predict sites and onset of co-translational interactions.

Overall, the manuscript provides a compelling story combining computational modelling and experimental validation. The computational methods appear solid, but I am not particularly familiar with these. The experimental section requires some clarifications and the manuscript as a whole a detailed proof-reading for clarity and consistency. In particular, several figure references appear to be inaccurate.

Detailed comments:

Page 2, left column, about middle: “We simulated the Cryo-EM structures of NatA and NatB” – I assume you modelled the structures of NatA and NatB based on the cryo-EM structures?

This line refers to the molecular dynamics simulations we ran on the cryo-EM structures. We thank the reviewer and edited for better clarity: “We ran the MD simulations on the cryo-EM structures of NatA and NatB.”

Page 4: “For NatA subunits, binding of TRiC/CCT was also hardly detected, interacting only with a very short segment in the catalytic subunit (residues 103-109)”. – Please show data as supplementary figure.

We thank the reviewer and have updated the manuscript accordingly. See revised Fig. S2, as well as revised Table S2 to include this: “SSb1/2 Chaperone binding sites of *S. cerevisiae* NatA and NatB subunits (data derived from accession code GSE93830) and TRiC/CCT (from Stein et al. (2019))”.

The NAA15 R354E/R355E double mutant has a similar effect as naa10delta, implying that NatA is effectively depleted. What happens to NAA10 in this mutant –is it soluble, or co-precipitating with in the aggregating NAA15 or is it degraded?

We appreciate the reviewer's insightful comment. Unfortunately, there was an unintentional error in our original submission: the N-terminomics results for the naa15Δ and the R354E, R355E mutants were mistakenly swapped. Consequently, we have corrected the order of the plots in Fig. 3i. Upon this correction it becomes evident that in the naa15Δ there is a significant decrease in the Nt-acetylation of NatA substrates, very similar to the reductions observed in naa10Δ. The R354E, R355E mutant exhibits a notable, albeit more subtle, reduction in acetylation:

Fig. 3i, Global N-terminome acetylation abundance ratios of Nt-acetylated peptides in the indicated mutants relative to the wildtype, plotted against their mass spectrometry (MS/MS) intensity – heavy plus light channels.

We have further tested Naa10 solubility in the Naa15 R354E, R355E strain, as well as the Naa15 R354A, R355A strain, and have not found significant changes compared to the wildtype:

Naa10 solubility is not significantly affected by Naa15's hotspots mutants: R354A, R355A; R354E, R355E.
(a,b) Imaging solubility analysis of Naa10-HA by immunostaining, with Anti-HA antibody conjugated to fluorescent Phycoerythrin (ab72479, abcam). Cells were grown and fixed as in Fig. 3e, permeabilized by Zymolyase and NP-40 treatment, then subjected to staining.
(a) Quantification of mean aggregates per cell as in Fig. 3f. $n > 200$.
(b) Representative images of the indicated strains.
(c) Sedimentation solubility analysis. Proteins were separated by centrifugation at 20,000g. All fractions (total, supernatant and pellet) were treated with 2% SDS and 4M urea for aggregates resuspension, then analyzed by immunoblotting against HA (rabbit).

Functional readout by N-terminome analysis:

Figure 3j: what is plotted on the x-axis: log 10 of the summed intensity of both channels?
Indeed, the X-axis refers to the Log10 of the total intensity of the heavy plus light channels. The figure legend was adjusted to clarify that.

Some NatA substrates are acetylated in the naa15delta. How is this rationalized? Are there specific subsets that require NAA15? What are the N-terminal sequence motifs of these populations?

We thank the reviewer for their comment. While both naa10Δ and naa15Δ demonstrate a substantial reduction in the acetylation of proteins' N-termini starting with Ser, Ala, or Thr (post Met1 excision), there are still (very) few N-terminal peptides beginning with these amino acids that remain unaffected by the knockouts. This observation aligns with findings reported in other proteomics studies, including:

1. "Proteome-derived peptide libraries allow detailed analysis of the substrate specificities of N(alpha)-acetyltransferases and point to hNaa10p as the post-translational actin N(alpha)-acetyltransferase", Van Damme et al., 2011
2. "The N-terminal Acetyltransferase Naa10/ARD1 Does Not Acetylate Lysine Residues", Magin et al., 2016

Similar findings were recently reported for NatC: “N-terminal acetylation shields proteins from degradation and promotes age-dependent motility and longevity” Varland et al., 2023, <https://www.nature.com/articles/s41467-023-42342-y> (Supplementary data 4).

In both mutants: Is this decrease in acetylation mirrored by an increase in the acetylated counterpart, or are these associated proteins degraded due to missing acetylation?

Yes, we do see in the tested strains a mirroring increase in the free N-termini that match NatA specificity. The peptides are derived from the open reading frame of the protein and they match tryptic specificity hence they are likely to represent the relative abundance of each proteoform (bubble plot below), and unlikely to be a degradation product. There is also a category of peptides that were exclusively observed in the acetylated form, hence their stability might be affected by the modification. We have now revised the manuscript to address this and added a detailed analysis of the peptides in this regard in Extended Data Table 2.

Venn diagram of Quantified-Only (QO), acetylated or free, N-terminal peptides of the Naa15 R354A, R355A mutant (left) and Naa15 R354E, R355E (right).

Bubble plot of free vs. acetylated quantified N-terminal peptides of the Naa15 R354A, R355A mutant (left) and Naa15 R354E, R355E (right). Included are only peptides that were found both free and acetylated. Peptides that show at least 2-fold increase in wildtype or the mutant strain were labelled as “WT” and “Mutant” respectively, and peptides that show a lower fold change were labelled as “Unchanged”. The size of each bubble is proportional to the number of peptides in each category.

Add a table summarizing your identifications with peptide sequence, modification, average fold change, significance score, and individual quantifications in each replicate.

We thank the reviewer and added the information in Extended Data Table 1.

Methods:

Enrichment of N-termini is not precise. Was Undecanal tagging really employed in the first step for modification, i.e. was undecanal used for positive selection of N-terminal peptides? *The schematic representation of the N-termini enrichment process in our study (Figure 3i) is accurate and, as depicted in the figure, it employs a negative selection strategy. The initial step of this enrichment (illustrated on the left side of the Fig. 3i) involves blocking all primary free amines on protein N-termini and lysine residues. This is achieved through dimethylation, using light labels for WT cells and heavy labels for KO/mutant cells. Subsequently, the samples are combined and subjected to trypsin digestion. Due to the prior*

blocking of lysine residues, trypsin functions similarly to ArgC, cleaving only at arginine residues. As a result, all peptides generated by trypsin digestion possess free N-termini, while the original N-terminal peptides retain their blocked N-termini. At this stage, undecanal is introduced. This compound selectively reacts with the trypsin-generated peptides (i.e., the internal peptides), as these are the only ones with free primary amines. Finally, the peptide mixture is applied to a reverse-phase column. Here, the original N-terminal peptides, which do not have the hydrophobic tag, are eluted with 60% acetonitrile. In contrast, the internal peptides, now tagged with the hydrophobic undecanal, remain bound to the column. We have now expanded the methods section to better clarify this. Please see the details in the answer below.

Similarly, the methodology used and cutoffs applied for data analysis must be better defined. Extend this section to match information provided in repository upload. **We thank the reviewer and revised the methods section accordingly. Please see revised text:**

“Sample Processing Protocol. Wildtype, naa10Δ, naa15Δ, mutant NAA15 (R354E, R355E), and mutant NAA15 (R354A, R355A) yeast cells were grown in appropriate SD media and plates to allow the selective growth of each strain. Single colonies of each strain were streaked onto 5 mL YPD media and grown overnight at 30 °C, 250 rpm. Cultures were then diluted 25-fold in 50 mL fresh YPD and grown until OD₅₉₅=1. Cells were harvested by centrifugation at 4000 rpm for 5 min in room temperature. Yeast pellets were washed 3 times with ice-cold water and flash-frozen in liquid nitrogen and stored in -80 °C. Frozen yeast pellets were dissolved in 250 μL 1% sodium deoxycholate in 100 mM HEPES pH-8.0 and incubated at 95 °C for 10 min. Protein concentration was determined using the BCA method. The samples were then subjected to denaturation, reduction, and alkylation processes. The naa10Δ, naa15Δ, and NAA15 mutant samples were labeled with light formaldehyde (C₁₂H₂) and WT samples labeled with heavy formaldehyde (C₁₃D₂). Finally, equal protein amounts of light-labeled NAA knockout/mutant and heavy-labeled wildtype samples were mixed and subjected to N-terminal enrichment based on HYTANE method using the protocol described in Hanna et al⁵⁹.

Following enrichment and desalting, the samples were analyzed by a tandem mass spectrometry using Thermo Q-Exactive Plus Orbitrap coupled to Easy nano-LC 1000 capillary HPLC. Enriched Terminal peptides were resolved on a homemade reverse phase capillary 30 cm long and 75 μm diameter, packed with 3.5 μm silica using ReproSil-Pur C18-AQ resin (Dr. Maisch GmbH). Elution was performed with a 120 min linear gradient of acetonitrile 5-28% (in 0.1% formic acid), followed by a 15-minute wash with 95% acetonitrile (in 0.1% formic acid), with all flow rates set to 0.15 μL/min. MS was conducted in a data-dependent acquisition mode for positive ions in an m/z range of 300-1500, with 70,000 resolution for MS1 and 17,500 resolution for MS2. The ten most dominant ions selected from the MS1 scan were fragmented using high-energy collisional dissociation with 25 normalized collision energy.

Data Processing Protocol. Data analysis was performed using the Trans Proteomic Pipeline (version 6.1)⁹⁴. Peptide search was done using Comet 2023.01 rev. 295. The rset of the analysis was conducted for only N-terminal acetylated peptides as described in the data analysis section of the protocol in Hanna et al⁵⁹.

Peptide scoring was done using PeptideProphet⁹⁶ at a false discovery rate of 1%. All data were searched against Uniprot Yeast proteome (UP000002311) containing 6090 sequences (downloaded April 2023) supplemented with known contaminant sequences and decoy sequences.”

Raw data is accessible, but only via Nature Communications Reporting summary file. Add

repository dataset identifier and reviewer access information to the methods section as standard practice for proteomeXchange-deposited data.

We thank the reviewer and have uploaded all the data to ProteomeXchange. Data are now available with the identifier PXD048082. The data will be made public upon publication. We added additional information to the method section: “Data are available via ProteomeXchange with identifier PXD048082. (Username: reviewer_pxd048082@ebi.ac.uk. Password: FARBSTgL)”.

References to figures:

Page 3: left column, about middle: A section mentions and discusses findings in the crystal structure with reference to Fig. 3g – this does not appear to match the information provided in the actual Fig. 3g.

We thank the reviewer and corrected the manuscript accordingly. Fig. 2g,h is now referred to instead of Fig. 3g.

Figure 3 c/d, references to this figure in page 4 top left paragraph: Fig 3.c and d appear changed (or panels swapped)

We thank the reviewer yet we did not find any inconsistencies in the order of the references or the panels. We have revised the legend text for clarity.

Page 5, discussion of N-termini: check reference to figure panels (3i referring to ratios instead of scheme, no reference to 3j)

We corrected this mistake and now both panels are properly referenced.

References 6 and 29 are identical.

We thank the reviewer and merged the two references.

REVIEWERS' COMMENTS

Reviewer #2 (Remarks to the Author):

The authors have addressed my concerns in a satisfactory manner.

Reviewer #3 (Remarks to the Author):

In their revision of the manuscript (NCOMMS-23-48035) authors have satisfactorily addressed all my comments and suggestions. The manuscript represents a novel, significant, and insightful contribution that advances the field. It will be of great interest to the readers of Nature Communications and can be published as is.

Reviewer #4 (Remarks to the Author):

The authors have addressed fully and adequately addressed my comments and concerns. Results and methods have been further clarified, inaccuracies in Figure references have been corrected and additional explanations added.

Congratulations on a fascinating story.